# NEURFLOW: INTERPRETING NEURAL NETWORKS THROUGH NEURON GROUPS AND FUNCTIONAL INTERACTIONS

**Tue M. Cao**[1]    **Nhat X. Hoang**[2]    **Hieu H. Pham**[3]    **Phi Le Nguyen**[1*]    **My T. Thai**[2*]

[1] Institute for AI Innovation and Societal Impact (AI4LIFE), Hanoi University of Science and Technology, Hanoi, Vietnam (`tue.cm210908@sis.hust.edu.vn`, `lenp@soict.hust.edu.vn`)
[2] University of Florida, Gainesville, Florida, USA {`hoangx, mythai`}`@ufl.edu`
[3] College of Engineering & Computer Science, VinUniversity, Hanoi, Vietnam
VinUni-Illinois Smart Health Center, VinUniversity, Hanoi, Vietnam
(`hieu.ph@vinuni.edu.vn`)

## ABSTRACT

Understanding the inner workings of neural networks is essential for enhancing model performance and interpretability. Current research predominantly focuses on examining the connection between individual neurons and the model's final predictions, which suffers from challenges in interpreting the internal workings of the model, particularly when neurons encode multiple unrelated features. In this paper, we propose a novel framework that transitions the focus from analyzing individual neurons to investigating groups of neurons, shifting the emphasis from neuron-output relationships to the functional interactions between neurons. Our automated framework, NeurFlow, first identifies core neurons and clusters them into groups based on shared functional relationships, enabling a more coherent and interpretable view of the network's internal processes. This approach facilitates the construction of a hierarchical circuit representing neuron interactions across layers, thus improving interpretability while reducing computational costs. Our extensive empirical studies validate the fidelity of our proposed NeurFlow. Additionally, we showcase its utility in practical applications such as image debugging and automatic concept labeling, thereby highlighting its potential to advance the field of neural network explainability. [1]

## 1 INTRODUCTION

The explainable AI (XAI) field has seen significant advancement in understanding the mechanisms of deep neural networks (DNNs). This field emerges from the growing need in decoding the internal representations, in hope of reverse engineering deep models into human interpretable program. Prior works have initiated on breaking down convolutional neural networks (CNNs) into interpretable neurons, understanding the models in the most fundamental units (Nguyen et al., 2016; Zeiler & Fergus, 2014; O'Mahony et al., 2023; Bykov et al., 2024). Extending further, one can examine the relation between neurons to gain insights on how the model works, within one layer (Vu et al., 2022), and between multiple layers (Cammarata et al., 2020). Ultimately, recent works try to generate circuits (Cammarata et al., 2020; Bykov et al., 2024; Wang et al., 2022c; Conmy et al., 2023) that create exhaustive explanations of how features are processed and evolve throughout the model.

The majority of existing methods focuses on individual neurons Oikarinen & Weng (2024b); La Rosa et al. (2023a) and their relationship to the model's final predictions Ghorbani & Zou (2020b); Wang et al. (2022b); Ghorbani & Zou (2020a), while giving less attention to exploring

---

[*]Corresponding Authors
[1]Source code: https://github.com/tue147/neurflow

and quantifying the relationships and interactions between neurons across different layers. These approaches are not only constrained by scalability challenges arising from the extensive number of neurons, but they also hinder a comprehensive understanding of the underlying mechanisms of DNNs. A notable example is the polysemantic phenomenon Mu & Andreas (2020); O'Mahony et al. (2023); Olah et al. (2020), where a single neuron is activated by several unrelated concepts. This phenomenon complicates the task of associating each neuron with a distinct feature and hampers the interpretation of how a model processes concepts based on the relationships among neurons. Drawing inspiration from human inference, which synthesizes information from a variety of sources, we contend that, in addition to individual neuron encoding multiple concepts (as demonstrated in prior studies O'Mahony et al. (2023); Olah et al. (2020)), groups of neurons within each layer also collectively encode the same concept. Furthermore, the decision-making process in neural networks is shaped not solely by the interactions between individual neurons, but rather by interactions among neuron groups.

This study seeks to explore the roles and interactions of neuron groups in shaping and developing concepts, enabling the execution of specific tasks. Due to the complex connections between large number of neurons, identifying those functions and there interactions is a daunting task. To overcome this, we demonstrate that for a particular task, only a subset of neurons—referred to as *core concept neurons*—play a crucial role as influential and concept-defining elements in neural networks. These neurons, when deactivated, significantly alter the associated concepts.

Focusing on core concept neurons allows us to view the intricate network in a simplified way, revealing the most important interactions between the groups of neurons. Therefore, we propose Neur-Flow framework that (1) identifies core concept neurons, (2) clusters these neurons into groups, and (3) investigates the functions and interactions of these groups. To enhance interpretability, we represent each neuron group by the set of visual features it encodes (i.e., named as neuron group's concept). Focusing on classification models, we construct, for each class of interest, a hierarchical tree in which nodes represent neuron groups (defined by the concepts they encode), and edge weights quantify the interactions between these groups.

Our key contributions are summarized as follows:

i) We introduce an innovative framework that systematically builds a circuit to elucidate the mechanisms by which core concept neuron groups operate and interact to achieve specific tasks. This entire process is automated, necessitating no human intervention or predefined concept labels. To our knowledge, we are the first to employ neuron groups as the fundamental units for explaining the internal workings of deep neural networks.

ii) We perform empirical studies to validate the proposed framework, demonstrating the optimality and fidelity of core concept neurons, and the reliability of interaction weights between core concept neuron groups.

iii) We provide experimental evidence showing that our framework can be applied to various tasks, including image debugging and automatic neuron concept labeling. Specifically, we confirm the biases found by Kim et al. (2024) on ImageNet (Russakovsky et al., 2015), which have not been proven, by masking the core concept neurons related to the biased features.

## 2 RELATED WORK

In an effort to understand the inner mechanism of DNNs, several branches of research have emerged:

**Concept based.** Kim et al. (2018) show that a model can be rigorously understood by assigning meaning to the activations, referred to as concept activation vectors. Subsequent works (Ghorbani et al., 2019; Zhang et al., 2021) have explored more complex methods for extracting these meanings, however, the relationships between concepts remain understudied. Fel et al. (2023); Kowal et al. (2024) address this limitation by constructing a graph of concepts with edges that quantify the relations. Their main intention is to see the evolution of concepts throughout the network layers. Nevertheless, they are unable to explain which parts of the model are responsible for these concepts.

**Neuron based.** Nguyen et al. (2016); Mu & Andreas (2020); O'Mahony et al. (2023); Bykov et al. (2024) invest effort in studying the meaning of neurons, in parallel, Vu et al. (2022); Ghorbani & Zou (2020c); Khakzar et al. (2021b) propose different approaches in identifying important neurons to the model output. These researches shed light on the function of individual neurons and their

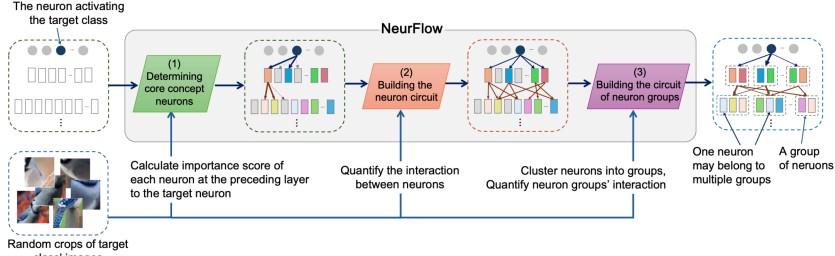

Figure 1: **Workflow of NeurFlow**, consisting of three main components: identifying core concept neurons in each layer, building the neuron circuit, and constructing the circuit of neuron groups.

impact on the prediction of the model. Recently, (Cammarata et al., 2020; Bykov et al., 2024; Achtibat et al., 2023) connect the neurons to form circuits that explain the behavior of a model throughout the layers, nevertheless, the circuits are constructed manually. Furthermore, a major limitation of all previous works is that they only analyze one neuron at a time. This approach is prone to the complex nature of neuron, namely polysemantic neurons, where neurons may encode multiple distinct features, making model interpretation via neurons challenging (Cammarata et al., 2020; O'Mahony et al., 2023). Lastly, Wang et al. (2022a); Kalibhat et al. (2023) find the group of neurons that encode the same concept, however, the relations among the groups and the influence of a group on the model's outputs are left unexplored.

**Graph based.** Ren et al. (2023); Zhou et al. (2024) try to approximate the mechanism of a model by considering the causal relations between the inputs and outputs. Another notable method (Zhang et al., 2018) generates a graph that highlights what visual features activate a feature map, for multiple layers. While this approach can be modified to form a circuit, the graph lacks meaningful edge weights. Consequently, it cannot quantify the contribution of each CNNs component to others and to the final prediction, unable to explain the inner mechanism (They use and-or-graph (Zhang et al., 2017) to form relations between components. However, unlike circuit, this new graph disregards the original structure of the model, where "concepts" of the first convolution could interact directly with "concepts" of the last convolution.). Subsequent work (Zhang et al., 2019) fixes this issue by building a decision tree to quantify the contribution of each feature map to the final predictions.

Our work aligns the most with explaining neurons and forming circuits. We address the common limitations of manual neuron labeling and circuit construction. We also propose a way to look at neurons not individually but in groups to overcome the common problem of polysemantic neurons. Additionally, we prioritize exploring the interactions between neuron groups across layers rather than focusing solely on the relationship between individual neurons and the model's output. Table 2 in Appendix B provides a comparison of our method with the most relevant existing studies.

## 3 NEURFLOW FRAMEWORK

### 3.1 PROBLEM FORMULATION

Our goal is to explain the internal mechanisms of deep neural networks (DNNs) by investigating how groups of neurons function and interact to encapsulate concepts, thereby performing a specific task. In particular, we focus on the classification problem, exploring how groups of neurons process visual features to identify a class. Given the exponential number of possible neuron groups, we focus only on core concept neurons. In addition, to facilitate human interpretation, we group these neurons through the common visual features they encode. In essence, we propose a comprehensive framework to address the following questions: *(i) Which neurons play a crucial role in each layer? (ii) How can these neurons be clustered, and what visual features does each neuron group encapsulate? (iii) How do groups of neurons in adjacent layers interact?*

Our problem can be formulated as follows: Given a pretrained classification network $F$ and a dataset $\mathcal{D}_c$ composed of exemplars from a specific class $c$, the goal is to construct a hierarchical tree whose vertices represent groups of core concept neurons in each network layer, and the edges capture the relationships between these groups. Figure 1 illustrates the workflow of our framework which comprises the following key components: (1) identifying core concept neurons (Section 3.3), (2) determining inter-layer relationships among neurons (Section 3.4), (3) clustering the core concept neurons into groups, and analyzing the interactions between these neuron groups (Section 3.5).

## 3.2 Definitions and Notations

In this paper, the term *neuron* refers to either a unit in a linear layer or a feature map in a convolutional layer. As suggested by Cammarata et al. (2020); Bykov et al. (2024); O'Mahony et al. (2023), each neuron is selectively activated by a distinct set of visual features, and by interpreting the neuron as a representation of these features, we can gain insights into the internal representations of a DNN. We refer to these visual features as the concept of the neuron. In the following definitions, let $a$ represent an arbitrary neuron located in layer $l$ of the pretrained network $F$. In this study, we do not rely on any predefined concepts. Instead, we enhance the original dataset $\mathcal{D}_c$ by cutting it into smaller patches with varying sizes. These patches serve as visual features for probing the model. We refer to this augmented dataset as $\mathcal{D}$, and denote $v$ as an arbitrary element of $\mathcal{D}$.

**Definition 1** (Neuron Concept). The neuron concept $\mathcal{V}_a$ of neuron $a$ is defined as the set of the top-$k$ image patches[1] that most strongly activate neuron $a$. Formally, the neuron concept of $a$ is expressed as $\mathcal{V}_a := \underset{\mathcal{V} \subset \mathcal{D}; |\mathcal{V}|=k}{\arg\max} \sum_{v \in \mathcal{V}} \phi_a(v)$, where $\phi_a(v) : \mathbb{D} \to \mathbb{R}$ represents the activation of neuron $a$ for a given input $v \in \mathcal{D}$, and $k$ is a hyperparameter.

An empirical analysis of the impact of $k$ (Appendix D.7) reveals that NeurFlow's performance is relatively insensitive to the selection of $k$.

**Definition 2** (Neuron Concept with Knockout). Let $M$ be the computational graph of the network $F$, $S$ be an arbitrary subset neurons of $M$, and $M \setminus S$ be the sub-graph of $M$ after removing $S$; let $\phi_a^{\overline{S}}$ be the activation of neuron $a$ computed from $M \setminus S$. The neuron concept of $a$ when knocking-out $S$ (denoted as $\mathcal{V}_a^{\overline{S}}$) is defined as $\mathcal{V}_a^{\overline{S}} := \underset{\mathcal{V} \subset \mathcal{D}; |\mathcal{V}|=k}{\arg\max} \sum_{v \in \mathcal{V}} \phi_a^{\overline{S}}(v)$.

We hypothesize that for each neuron $a$, only a small subset of neurons from the preceding layer exert the most significant influence on $a$. In particular, knocking out these neurons would lead to a substantial change in the concept associated with $a$. We refer to these neurons as *core concept neurons* and provide a formal definition in the following.

**Definition 3** (Core Concept Neurons). Given a neuron $a$ at layer $l$, core concept neurons of $a$ (denoted as $\mathbb{S}_a$) is the sub-set of neurons at the previous layer $l - 1$ satisfying the following conditions:

$$\mathbb{S}_a := \underset{S \subseteq \mathbb{S}; |S| \leq \tau}{\arg\min} \left| \mathcal{V}_a^{\overline{S}} \cap \mathcal{V}_a \right|, \tag{1}$$

where $\mathbb{S}$ is set of all neurons at layer $l - 1$ and $\tau$ is a predefined threshold. Intuitively, the core concept neurons for a target neuron $a$ are those that play an important role in defining the concepts represented by $a$. In practice, the value of $\tau$ may vary across the network layers, its impact will be elaborated upon in Sections 4.

In the following, we denote by $\phi^{1,l-1}(v) : \mathbb{D} \to \mathbb{R}^{m \times w \times h}$ the function that maps the input $v$ to the feature maps at the $(l-1)$-th layer of the model, where $m$ represents the number of channels, and $w \times h$ indicates the dimensions of each feature map. Furthermore, we adopt the notation $|.|$ to indicate the cardinality of a set, while $\|.\|$ is employed to represent the absolute value. We summarize all the notations in Table 1 (Appendix A).

## 3.3 Identifying core concept Neurons

Given a neuron $a$, we describe our algorithm for identifying its core concept neuron set $\mathbb{S}_a$. This process consists of two main steps: determining $a$'s concept $\mathcal{V}_a$ according to Definition 1, and identifying core concept neurons following Definition 3.

Firstly, we generate a set of image patches $\mathcal{D}$ by augmenting the original dataset $\mathcal{D}_c$, which consists of images that the model classifies as class $c$. Since neurons can detect visual features at different levels of granularity, we divide each image in $\mathcal{D}_c$ into smaller patches using various crop sizes, where smaller patches capture simpler visual features and larger patches represent more complex ones. We subsequently evaluate all items in $\mathcal{D}$ to identify the top-$k$ image patches that induce the highest activation in neuron $a$, thereby constructing $\mathcal{V}_a$.

---

[1] each image patch is a piece cropped from image set.

With $\mathcal{V}_a$ identified, one could determine the core concept neurons through a brute-force search over all possible candidates. However, this naive approach is computationally infeasible. To this end, we define a metric named *importance score* that quantifies the attribution of a neuron $s_i$ to $a$. The importance score can be intuitively seen as integrated gradients (Sundararajan et al. (2017)) of $a$ to $s_i$ calculated across all elements of $\mathcal{V}_a$, calculated as follows:

$$T(a, s_i, \mathcal{V}_a) = \sum_{v \in \mathcal{V}_a} \sum_{\substack{x \in \phi_{s_i}^{1,l-1}(v); \\ y \in \phi_a^{l-1,l}(\phi^{1,l-1}(v))}} x \times \frac{1}{N} \left( \sum_{n=1}^{N} \frac{\partial y(\frac{n}{N}x)}{\partial x} \right), \tag{2}$$

where $\phi_{s_i}^{1,l-1}$ is the element of $\phi^{1,l-1}$ corresponding to neuron $s_i$, $\phi_a^{l-1,l}$ depicts the function mapping from the activation vector of layer $l-1$ to the activation of neuron $a$, and $N$ is the step size. Utilizing the *importance scores* of all neurons in the preceding layer, the set of core concept neurons is identified by selecting the top $\tau$ neurons that exhibit the highest absolute scores. To justify the use of integrated gradients, we empirically show a strong correlation between the absolute values of $T(a, s_i, \mathcal{V}_a)$ and the change in $a$'s concept after knocking out $s_i$, as demonstrated in Section 4. Additionally, we compare our method with other attribution techniques in Appendix D.1.

## 3.4 CONSTRUCTING CORE CONCEPT NEURON CIRCUIT

For each class of interest $c$, the neuron circuit $\mathcal{H}_c$ is represented as a *hierarchical hypertree*[2], with the root $a_c$ being the neuron in the logit layer (ouput) associated with class $c$. The nodes in each layer of the tree $\mathcal{H}_c$ are the core concept neurons of those in the layer above, and branches connecting a parent node $a$ and its child $s_i \in \mathbb{S}_a$ represents the contributions of $s_i$ to $a$'s concept.

As mentioned in (Cammarata et al., 2020; O'Mahony et al., 2023), neurons often exhibit polysemantic behavior, meaning that a single neuron may encode multiple distinct visual features. In other words, the visual features within a concept $\mathcal{V}_a$ of neuron $a$ may not share the same meaning and can be categorized into distinct groups, which we term *semantic groups*. We hypothesize that each core concept neuron $s_i$ makes a distinct contribution to each semantic group of neuron $a$. To model this relationship, we represent the interaction between $s_i$ and $a$ through multiple connections, where the $j$-th connection reflects $s_i$'s influence on $\mathcal{V}_{a,j}$, the $j$-th semantic group of $a$.

At a conceptual level, the algorithm for constructing the hypertree $\mathcal{H}_c$ proceeds through the following steps: (1) employing our core concept neuron identification algorithm to determine the children of each node in the tree (Section 3.3), (2) clustering the neuron concept of each parent node into semantic groups, and (3) assigning weights to each branch connecting a child node to the semantic groups of its parent. Figure 2 illustrates our algorithm. The complete algorithm for constructing the core concept neuron circuit is presented in Appendix E. We provide a detailed explanation of these steps below.

**Determining semantic groups.** Let the concept $\mathcal{V}_a$ corresponding to $a$ be composed of $k$ elements $\{v_a^1, \ldots, v_a^k\}$. For each visual feature $v_a^i$ ($i = 1, \ldots, k$), we define its representative vector $r(v_a^i) \in \mathbb{R}^m$ as:

$$r(v_a^i) = \left[ \texttt{mean}\left(\phi_1^{1,l-1}(v_a^i)\right), \ldots, \texttt{mean}\left(\phi_m^{1,l-1}(v_a^i)\right) \right], \tag{3}$$

where $\phi_j^{1,l-1}(v_a^i)$ ($j = 1, \ldots, m$) represents the $j$-th feature map and $\texttt{mean}\left(\phi_j^{1,l-1}(v_a^i)\right)$ denotes the average value across its all elements. Next, we use agglomerative clustering (Murtagh & Legendre, 2014) to divide the set $\{v_a^1, \ldots, v_a^k\}$ into clusters, where the distance between two visual features $v_a^p, v_a^q$ is defined by the distance between their corresponding representative vectors $r(v_a^p)$, $r(v_a^q)$. The Silhouette score (Rousseeuw, 1987) is employed to ascertain the optimal number of clusters. The complete procedure is detailed in Algorithm 2.

**Calculating edge weight.** The weight $w(a, s_i, \mathcal{V}_{a,j})$ of the branch connecting a child $s_i$ and its parent $a$'s $j$-th semantic group $\mathcal{V}_{a,j}$ is defined as:

$$w(a, s_i, \mathcal{V}_{a,j}) = \frac{T(a, s_i, \mathcal{V}_{a,j})}{\sum_{s \in \mathbb{S}_a} \|T(a, s, \mathcal{V}_{a,j})\|}, \tag{4}$$

where $T(a, s_i, \mathcal{V}_{a,j})$ is the importance score of $s_i$ to $a$ calculated over $\mathcal{V}_{a,j}$.

---

[2]A hypertree is a tree in which each child-parent pair may be connected by multiple edges.

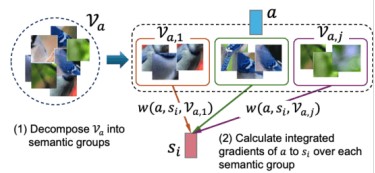
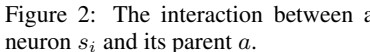
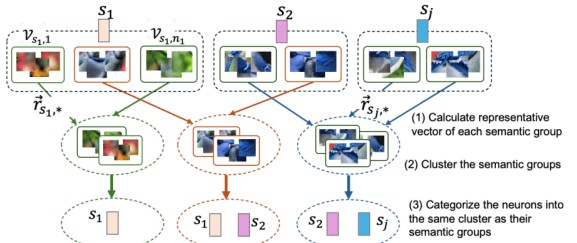

Figure 2: The interaction between a neuron $s_i$ and its parent $a$.

Figure 3: Illustration of our algorithm to determine groups of neurons.

### 3.5 DETERMINING GROUPS OF NEURONS AND CONSTRUCTING CONCEPT CIRCUIT

This section describes our algorithms to (1) cluster the set of core concept neurons $\mathbb{S}_a = \{s_1, ..., s_k\}$ into distinct groups, (2) identifying the concept associated with each group, and (3) quantifying the interaction between the groups.

**Clustering neurons into groups.** As mentioned in the previous section, a single neuron can encode multiple distinct visual features, while several neurons may also capture the same visual feature Cammarata et al. (2020). We hypothesize that, due to the polysemantic nature of neurons (Cammarata et al., 2020; O'Mahony et al., 2023), a model may struggle to accurately determine whether a concept is present in an input image by relying on a single neuron. As a result, the model processes visual features not by considering individual neurons in isolation but rather by operating at the level of neuron groups. Intuitively, a group of neurons consists of those that capture similar visual features. This can be interpreted as *two neurons belonging to the same group if they share similar semantic concept groups*.

Building on this intuition, we develop a neuron clustering algorithm based on the semantic groups of each neuron's concept (Figure 3). Specifically, let $\mathcal{V}_{s_i}$ represent the concept of neuron $s_i$ (i.e., the primary visual features it encodes), which can be decomposed into several semantic groups $\{\mathcal{V}_{s_i,1}, ..., \mathcal{V}_{s_i,n_i}\}$ (see Section 3.4), where $n_i$ is the number of semantic groups encoded by $s_i$. For each semantic group $\mathcal{V}_{s_i,j}$, we calculate its representative activation vector $\overrightarrow{r_{s_i,j}}$ by averaging the feature maps of all its visual features, i.e., $\overrightarrow{r_{s_i,j}} := \frac{1}{|\mathcal{V}_{s,j}|} \sum_{v_s \in \mathcal{V}_{s,j}} mean(\phi^{1,l-1}(v_s))$. We then apply the agglomerative clustering algorithm to group the semantic groups $\mathcal{V}_{s_i,j}$ ($i = 1, ..., k; j = 1, ..., n_i$), where the distance between any two groups $\mathcal{V}_{s_i,u}$ and $\mathcal{V}_{s_j,w}$ is determined by the distance of their respective representative activation vectors, $\overrightarrow{r_{s_i,u}}$ and $\overrightarrow{r_{s_j,w}}$. Finally, we assign neurons $s_1, ..., s_k$ to the same groups based on their semantic concept groups. Specifically, neurons $s_i$ and $s_j$ are clustered together if there exists a semantic group $\mathcal{V}_{s_i,u}$ (of $s_i$) and a semantic group $\mathcal{V}_{s_j,w}$ (of $s_j$) belonging to the same group.

**Finding neuron group concept automatically.** We define the concept associated with a group of neurons as the union of all visual features from the corresponding semantic groups. Specifically, let $\{\mathcal{V}_{G,1}, ..., \mathcal{V}_{G,k}\}$ represent the semantic groups categorized into a cluster, with their corresponding neurons $\{s_{G,1}, ..., s_{G,k}\}$ grouped together in the same set, denoted as $G$. The concept of this group, denoted as $\mathbb{V}_G$, is then defined as the union of the sets $\{\mathcal{V}_{G,1}, ..., \mathcal{V}_{G,k}\}$, i.e., $\mathbb{V}_G := \bigcup_{i=1}^{k} \mathcal{V}_{G,i}$. We leverage a Multimodal LLM to automatically assign labels to the concept, thereby eliminating the need for a predefined labeled concept dictionary. Further details on the design of the prompts are provided in the Appendix G.

**Constructing concept circuit.** For a given class $c$, the concept circuit $\mathcal{C}_c$ is a hierarchical tree where each node represents a neuron group concept (*NGC*), and each edge illustrates the contribution of the child neuron group to its parent. For a node $G$, we denote by $\mathbb{V}_G = \{\mathcal{V}_{G,1}, ..., \mathcal{V}_{G,|\mathbb{V}_G|}\}$ the set of semantic groups associated with $G$, and $\mathbb{S}_G = \{s_{G,1}, ..., s_{G,|\mathbb{S}_G|}\}$ represent the neurons corresponding to the semantic groups in $\mathbb{V}_G$, i.e., $s_{G,j}$ is the core concept neuron possesses the semantic group $\mathcal{V}_{G,j}$ ($j = 1, ..., |\mathbb{V}_G|$). Let $G_i$ and $G_j$ be a child-parent pair in the tree, then, the relationship between $G_i$ and $G_j$ (quantified by $W(G_i, G_j)$) is represented by two aspects: the number of edges connecting elements of $G_i$ and $G_j$, and the weights of those connecting edges. The more the edges and the higher the edge weights, the stronger the relationship between $G_i$ and $G_j$. Accordingly, we define the weight of branch connecting a child $G_i$ to its parent $G_j$ as sum of the attribution of each neuron in $\mathbb{S}_{G_i}$ with each neuron in $\mathbb{S}_{G_j}$: $W(G_i, G_j) := \sum_{\substack{s_{G_i,q} \in \mathbb{S}_{G_i} \\ s_{G_j,p} \in \mathbb{S}_{G_j}}} w(s_{G_j,p}, s_{G_i,q}, \mathcal{V}_{G_j,p})$.

## 4 EXPERIMENTAL EVALUATION

We perform an extensive empirical study to investigating three aspects, including: optimality of core concept neurons, fidelity of core concept neurons, and fidelity of neuron interaction weights. Our experiments are performed on ResNet50 (He et al., 2016) and GoogLeNet Szegedy et al. (2015) using the ILSVRC2012 validation set (Russakovsky et al., 2015). The models are pretrained in Pytorch (Paszke et al., 2019), and layer names follow Pytorch's conventions (e.g., *layer4.2* for ResNet50). Unless otherwise specified, the input parameters are $\tau = 16$, $N = 50$, and $k = 50$, where the top 50 images with the highest activation on the target neuron are considered as its concept. We will release the source code once the paper is published.

**Optimality of core concept neurons.** According to Definition 3, the core concept neuron set $\mathbb{S}_a$ for neuron $a$ is the set that minimizes the objective function $\left| \mathcal{V}_a^{\overline{\mathbb{S}_a}} \cap \mathcal{V}_a \right|$ without exceeding the cardinality $\tau$. To evaluate our heuristic solution, we define a loss function $\mathcal{L}(S, a) := \left| \mathcal{V}_a^{\overline{S}} \cap \mathcal{V}_a \right| / |\mathcal{V}_a|$, balancing these two objectives. In this experiment, our objective is to demonstrate that the core concept neurons, $\mathbb{S}_a$, identified by our algorithm are near-optimal. Ideally, a brute-force search over all possible combinations of $\tau$

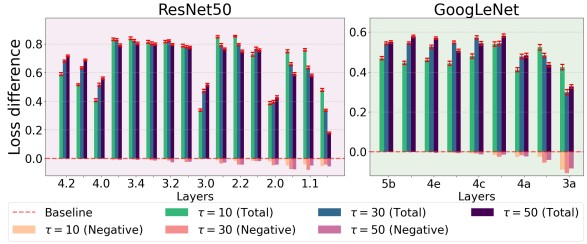

Figure 4: **The difference in losses between core concept neurons and random neuron combinations.** The blue-toned bars represent the average losses, while the pink-toned bars indicate instances where random neuron combinations result in smaller losses compared to core concept neurons.

neurons would be conducted to demonstrate that these combinations yield a higher loss function value compared to $\mathbb{S}_a$. However, such an approach is computationally infeasible due to its prohibitive cost. Consequently, we perform experiments using a large set of randomly selected combinations. Specifically, we use three different values of $\tau$, specifically $10, 30, 50$. For each setting, we randomly select 50 target neurons (denoted by $a_i$) from 10 distinct classes (five neurons for each class). For each target neuron $a_i$, we determine its core concept neuron set $\mathbb{S}_{a_i}$ using our algorithm and generate 100 random neuron combinations, with the same cardinality as $\mathbb{S}_{a_i}$, from the preceding layer of $a_i$. In total, the experiments are performed over $15,000$ cases per layer for each model. We compare the loss differences between $\mathbb{S}_{a_i}$ and the random neuron combinations. These average differences along with 99% the confidence intervals are shown in Figure 4. Additionally, we report cases where the random combinations resulted in a smaller loss than our core concept neurons. As observed, the average differences are positive in all cases, indicating that replacing the core concept neurons identified by our algorithm with random ones generally leads to a significant increase in the loss for both models. Furthermore, only a few cases show a random combination achieving a smaller loss than our core concept neurons, and in those instances, the gap is negligible.

**Fidelity of core concept neurons.** We evaluate the impact of the identified core concept neurons on the model's performance by comparing two variants: (1) *Retaining version*–all neurons masked except for the core concept neurons, and (2) *Masking version* version–only the core concept neurons are masked. Intuitively, a higher performance in the *Retaining version* and a lower performance in the *Masking version* would indicate that the core concept neurons play a significant role in the model's performance. We compare the performance of these two versions against models obtained by performing retraining and masking on equal numbers of random neurons. We select 50 random classes and apply the retaining and masking operations at two levels: on a single layer or across multiple layers. In the multi-layer scenario, masking or retaining is applied from the linear classifier down to a specified layer. Figure 5 presents the results for $\tau = 4, 8$, and 16. The $y$-axis indicates changes in model accuracy, where a value of 1 implies that masking neurons does not affect predictions. It is evident that masking core concept neurons consistently results in a more pronounced decline in performance compared to masking random neuron combinations. Moreover, the rate of decline in accuracy, moving from higher to lower layers, is considerably steeper for the core concept neurons than for random neurons. The most significant discrepancy occurs at layer *5a* of the GoogLeNet model, where masking core concept neurons at this layer reduces model accuracy to nearly 0, while masking random neurons has a minimal effect on performance. *Retaining version*, preserving only the core concept neurons allows the model to maintain its performance substantially better than when random neurons are retained. This experiment also demonstrates that the value of

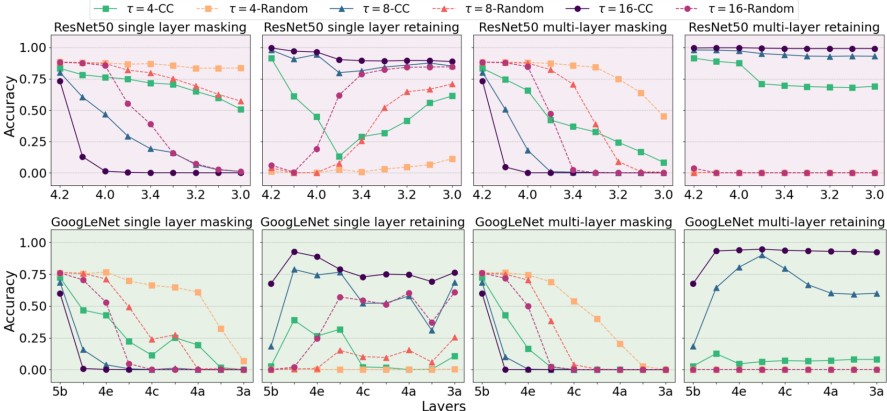

Figure 5: **Effects of neuron groups on model's performance.** Retaining only the core concept (denoted as CC) neurons preserves high accuracy, whereas masking them leads to a significant drop in performance. In contrast, random neuron combinations show the opposite trend.

$\tau$ represents a trade-off between the simplicity of the circuit and the comprehensiveness of capturing the core concept neurons. A smaller $\tau$ results in greater instability in the model's performance during the retaining experiment, leading to a more pronounced performance drop. For more discussion on the impacts of $\tau$, please refer to Appendix D.6.

We conduct an experiment to show that adding non-core neurons to the concept core neurons set identified by NeurFlow has minimal impact the model's performance. Specifically, we perform the Fidelity experiment with $\tau = 16$, incorporating $50\%$ more non-core neurons (i.e., those that are not concept core neurons), and evaluated their impact on model accuracy. These neurons were selected greedily, prioritizing those with the highest scores as ranked by NeuronMCT Khakzar et al. (2021a). The results in Figure 17 (Appendix D.6) indicate that for ResNet-50, adding non-core neurons had little to no effect on improving model performance, confirming that when $\tau$ is sufficiently large, our algorithm ensures completeness.

**Fidelity of neuron interaction weights.** The edge weight representing the interaction between core concept neurons (or groups of core concept neurons) is defined using Integrated Gradients (IG) (Definition 3). Without the ground truth, we evaluate the fidelity of edge weights based on the following rationale: if the weights assigned by our definition are meaningful, they should accurately rank the importance of neurons in the preceding layer in detecting the concept represented by a target neuron in the subsequent layer. We demonstrate that our IG-based scores exhibit a strong correlation with the loss, not only for single neuron setup but also for groups of neurons. Specifically, we randomly select 10 target neurons from 10 distinct classes (denoted as $a_i$, where $i = 1, \ldots, 10$). For each target neuron $a_i$, we generate random combinations of neurons from the preceding layer. We then measure the correlation between the losses caused by these random neuron combinations and the sum of the absolute values of their IG-based importance scores with respect to $a_i$. The experiments are conducted using 500 neuron combinations, with cardinality ($\tau$) varying from 1 to 50. Figure 6 presents the average correlation across all combinations. The results indicate that for $\tau < 50$, IG-based scores maintain a high correlation across all layers. Notably, for $\tau = 1$, the correlation consistently exceeds 0.6 in both models, and up to almost perfect correlations for several layers in ResNet50. While the correlation diminishes as $\tau$ increases, our focus is on a small subset of core concept; thus, for a sparse sub-graph of core concept neurons, these results are considered satisfactory. We further compare our defined IG-based score with other attribution methods, including the one used in Vu et al. (2022), in the Appendix D.1.

**Quantitative comparison of NeurFlow with existing approaches.** While our approach focuses on identifying core concept neurons relative to a specific target neuron, we demonstrate that the neurons identified by our method also significantly influence the model's final output. To validate this, we analyzed the overlap between our core concept neurons and the critical neurons identified by Vu et al. (2022), and NeuronMCT (Khakzar et al., 2021a). The $F_1$ scores

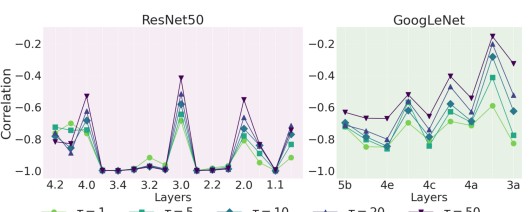

Figure 6: Correlation between loss and our defined IG-based importance scores.

for these overlaps are presented in Table 3 (Appendix D.4). The results indicate that NeurFlow iden-

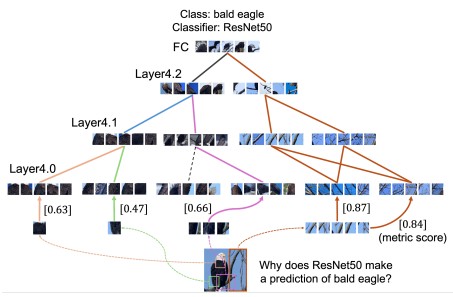

Figure 7: **Using NeurFlow to reveal the reason behind model's prediction.** The top concepts can be traced throughout the circuit.

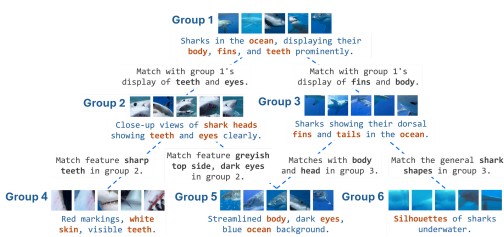

Figure 8: Demonstration for automatically labelling and explaining the relation of NGCs on class "great white shark" using GPT4-o (OpenAI, 2024). The captions and the names of the NGCs are highlighted in blue, while the relations are in black.

tifies core concept neurons largely similar to those found by NeuronMCT, even though it does not explicitly find critical neurons to the model's output. Additionally, we compare our approach for identifying core concept neurons for a specific target neuron with the method proposed in Cammarata et al. (2020). Details of this experiment can be found in Section D.5 (Appendix D.5). The results, summarized in Table 4 (Appendix), show that our method is more effective in identifying core concept neurons.

## 5 APPLICATIONS

We outlines some applications of NeurFlow. We hypothesize that, as one neuron can have multiple meanings, a DNN looks at a group of neurons rather than individually to determine the exact features of the input. Hence, we propose a metric that assesses a model's confidence in determining whether the input contains a specific visual feature. For a group $G$ with core concept neurons $\mathbb{S}_G = \{s_{G,1}, \ldots, s_{G,|\mathbb{S}_G|}\}$, the metric denoted as $M(v, \mathbb{S}_G, \mathcal{D}) = \exp(\frac{1}{|\mathbb{S}_G|} \sum_{s \in \mathbb{S}_G} \log(\|\phi_s(v)/\max(\phi_s, \mathcal{D}))\|)$, where $v \in \mathcal{D}$ and $\max(\phi_s, \mathcal{D})$ is the highest value of activation of neuron $s \in \mathbb{S}_G$ on dataset $\mathcal{D}$. This returns high score when all neurons in $G$ have high activation (indicating high confidence), while resulting in almost zero if any neuron in the group has low activation (indicating low confidence). We can use this metric to determine how similar the features in the input image are to the predetermined neuron groups concept. The specific setup can be found in the Appendix F. Figures 7 and 9 demonstrate the usage of the metric and the concept circuit. We use the term *NGC* to denote the concept of a neuron group.

### 5.1 IMAGE DEBUGGING

We aim to use the concept circuit to identify concepts contributing to false prediction, which we call *image debugging*. If a concept contributes to a class when it should not, we say that the prediction (or equivalently, the model) is *biased* by that concept. Kim et al. (2024) propose a framework for detecting biases in a vision model by generating captions for the predicted images and tracking the common keywords found in the captions. With this method, they concluded that the pretrained ResNet50 is biased by "flower pedals" in the class "bee". However, correlational features do not imply causation and can lead to misjudgments. We verify and enhance the causality of their claim by examining the concept circuit of class "bee", and conducting experiments on the probabilities of the final predictions with and without neurons that related to "flowers". Additionally, we discover that the model also suffers from "green background" bias (resemble "leaves"), which is not mentioned in Kim et al. (2024).

Figure 9 shows the process of debugging false positive images. Three different concepts are presented in *layer4.2* of ResNet50, representing "pink pedals", "green background", and "bee" respectively (we choose this layer as it has a small set of NGCs, however, our following experiment is consistent for multiple layers and with different classes). We discover that most of the false positive images have high metric score for "pedal" and "green background". To further verify the impact of these biased features, we mask all neurons in the groups of the respective concepts and find that the probability of the predictions are distorted drastically (and predictions is no longer "bee"), as opposed to masking random neurons, which yield negligible changes.

This implies the dependence on the biased concept. *But how do we know that the groups reflect the respective visual features?* If these groups indeed represent the visual features, then masking them should hinder the classification probability for images that include those features. We highlight the

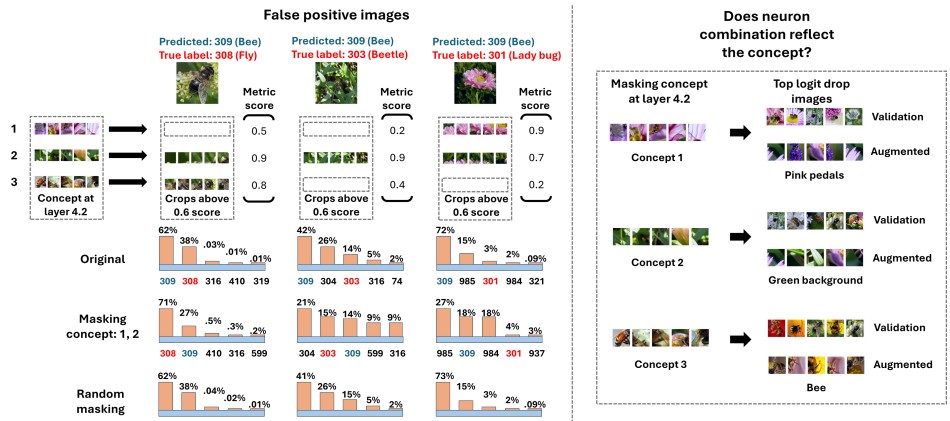

Figure 9: (left) The metric scores of false positive images for each concept in *layer4.2* of ResNet50. (right) Showing the images that have the greatest drop in the activation of the logit neuron when masking each group concept. Verifying that the neuron groups indeed reflect the concepts.

top images that have the largest decrease in the value of the logit neuron (corresponding to class "bee") on both validation set of the target class and augmented dataset (see Section 3.3). As shown in Figure 9, this process indeed yields the images that contain the respective features.

To demonstrate how NeurFlow's findings differ from those of existing methods, we conduct a qualitative experiment comparing the core concept neurons identified by NeurFlow with those identified by NeuCEPT Vu et al. (2022). Detailed information about this experiment is provided in Appendix D.3. Our observations indicate that the top logit drop images identified by NeurFlow align better with the representative examples of the corresponding concepts. Moreover, masking the core concept neuron groups identified by NeurFlow resulted in more significant changes to prediction probabilities while utilizing fewer neurons compared to the groups identified by NeuCEPT.

## 5.2 AUTOMATIC IDENTIFICATION OF LAYER-BY-LAYER RELATIONS

While automatically discovering concepts from inner representation has been a prominent field of research (Fel et al., 2023), automatically explaining the resulting concepts is often ignored, relying on manual annotations. Bykov et al. (2024) utilize label description in ImageNet dataset to generate caption for neurons, however, these annotations is limited and can not be used to label low level concepts. Drawing inspiration from Hoang-Xuan et al. (2024); Kalibhat et al. (2023), we go one step further and not only use MLLM to label the (group of) neurons but also explain the relations between them in consecutive layers. Thus, we show the prospect of completing the whole picture of abstracting and explaining the inner representation in a systematic manner.

Specifically, for two consecutive layers, we ask MLLM to describe the common visual features in a NGC, then matching with those of the top NGC (with the highest weights) at the preceding layer. This can be done iteratively throughout the concept circuit, generating a comprehensive explanation without human effort. We use a popular technique (Wei et al., 2022) to guide GPT4-o (OpenAI, 2024) step by step in captioning and in visual feature matching. Figure 8 shows an example of applying this technique to concept circuit of class "great white shark". We observe that MLLM can correctly identify the common visual features within exemplary images of NGCs. Furthermore, MLLM is able to match the features from lower level NGCs to those at higher level, detailing formation of new features, showing the potential of explaining in automation, capturing the gradual process of constructing the output of the model. The prompt used in this experiment is available in Appendix G.

## 6 CONCLUSION

We introduced NeurFlow, a framework that systematically elucidates the function and interactions of neuron groups within neural networks. By focusing on the most important neurons, we revealed relationships between neuron groups, which are often obscured by the inherent complexity of neural network structures. Furthermore, we fully automated the processes of identifying, interpreting, and annotating neuron group circuits using large language models. Our method aims to provide a more efficient and comprehensive approach to the automated interpretation of neural activity and applicability of NeurFlow across a variety of domains, including image debugging and automatic concept labeling.

ACKNOWLEDGMENTS

This work was funded by Vingroup Joint Stock Company (Vingroup JSC),Vingroup, and supported by Vingroup Innovation Foundation (VINIF) under project code VINIF.2021.DA00128.

This work is partially supported by the US National Science Foundation under SCH-2123809 project.

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

## A  NOTATIONS

We summarize the notations used in this work in Table 1.

Table 1: Notations summarization.

| Notation | Meaning |
|---|---|
| $a$ | A neuron in a model at layer $l+1$ |
| $S$ | A set of neurons at layer $l$ |
| $\mathcal{V}_a$ | Concept of $a$: the top-$k$ highest image patches that activate $a$ |
| $\mathcal{V}_a^{\overline{S}}$ | Concept of $a$ when knocking out $S$ |
| $\mathcal{V}_{a,j}$ | The $j$-th semantic group of $a$ |
| $\tau$ | The number of core concept neurons of $a$ |
| $\mathbb{S}_a$ | The set of core concept neurons of $a$ |
| $\phi^{1,l}$ | The function that maps from the dataset to the activation at layer $l$ of the model |
| $T(a, s_i, \mathcal{V}_a)$ | The importance score of $s_i \in \mathbb{S}_a$ w.r.t $a$ on $\mathcal{V}_a$ |
| $w(a, s_i, \mathcal{V}_{a,j})$ | The normalized importance score of $s_i$ w.r.t $a$ on $\mathcal{V}_{a,j}$ |
| $r(v)$ | The activation vector of an input $v$ |
| $\mathcal{V}_{s_i,j}$ | The representative activation vector of the $j$-th semantic group |
| $G$ | A neuron group |
| $\mathbb{S}_G$ | The set of neurons of $G$ |
| $\mathbb{V}_G$ | The concept of $G$ |
| $W(G_i, G_j)$ | The edge weight between $G_i$ and $G_j$ |

## B  RELATED WORKS SUMMARIZATION

Table 2 compares our proposed method and existing approaches.

Table 2: Comparison of NeurFlow and existing approaches.

| Method | Objectives | Level of granularity | Interaction quantification |
|---|---|---|---|
| Vu et al. (2022) | Finding critical neurons to the model's output | | |
| Ghorbani & Zou (2020c) | | | |
| Khakzar et al. (2021b) | | | |
| O'Mahony et al. (2023) | Individual neuron explanation | Neuron | N/A |
| Mu & Andreas (2020) | | | |
| La Rosa et al. (2023b) | | | |
| Oikarinen & Weng (2024a) | | | |
| Mu & Andreas (2020) | | | |
| Bykov et al. (2024) | | | |
| Kalibhat et al. (2023) | | Group of neurons | |
| Wang et al. (2022a) | | | |
| Kowal et al. (2024) | Determining concept connectivity | Concept | Concept interaction |
| **NeurFlow (Ours)** | Determining groups of neurons' function and interaction | Group of neurons | Neuron group interaction |

## C  LIMITATIONS AND DISCUSSION

While we view NeurFlow as a significant step toward understanding the function and interaction of neuron groups, it is not without limitations. Our approach defines the concept of neurons as the top-$k$ most activated visual features, a common practice in the field (O'Mahony et al., 2023; Mu & Andreas, 2020; Nguyen et al., 2016). However, other researchers have broadened this definition to include concepts spanning a wider range of activation patterns (La Rosa et al., 2023b; Oikarinen & Weng, 2024a). This limitation highlights a promising direction for future research: developing more

flexible frameworks that incorporate both top-$k$ activation and more distributed neural activation patterns.

Furthermore, our research primarily focus on CNNs, which follows the main focus of a range of previous works in the field (Cammarata et al., 2020; O'Mahony et al., 2023; Nguyen et al., 2016; Mu & Andreas, 2020). However, we can apply our framework onto different DNN architectures by following several steps: 1) define the granularity level of neurons (i.e. individual units, feature maps, attention heads etc.); 2) iteratively identify the target neuron concept and the core concept neurons; 3) cluster the core concept neurons into groups and construct the concept graph. While exploring the differences in the inner workings of various architectures is valuable, we leave this promising direction for future works.

## D ABLATION STUDIES

### D.1 COMPARISON OF ATTRIBUTION METHODS

In this section, we run an ablation study on different choices of attribution method apart form our integrated gradient (IG) approach, verifying that IG-based score is the most suitable for the quantification of edge weights. We assess four additional common pixel attribution methods, including LRP (Bach et al., 2015), Guided Backpropagation (Springenberg et al., 2014), SmoothGrad (Smilkov et al., 2017), Saliency (Simonyan et al., 2014), Gradient Shap (Lundberg, 2017). Notably, Smooth-Grad and Gradient Shap are a follow-up versions of IG. Furthermore, we also evaluate attribution method used in Vu et al. (2022), which also find important neurons and attributing scores to them, referred to as Knockoff (Candes et al., 2018). We run on the same setup as in Section 4 for $\tau$ ranging from 0 to 50. For easier comparison, we report the mean correlations of all values of $\tau$. Figure 11 show the mean correlations across the last 10 layers of ResNet50 (He et al., 2016) and GoogLeNet (Szegedy et al., 2015). The Integrated Gradient consistently yields higher correlations compared to other attribution method, surpassing its follow-up version SmoothGrad, while being comparable with Gradient Shap. Furthermore, Knockoff shows a poor performance in ranking the importance of neurons compared to other attribution methods.

Additionally, we also assess the running time of each method. Specifically, we recorded the run time of each method on 50 images on CPU (we implement Knockoff on KnockPy library (Spector & Janson, 2021+) which does not run on GPU, hence, we evaluate all others on CPU for a fair comparison) across all layers of GoogLeNet. The results in Figure 10 show that IG maintain a small running time compared to the follow-up method (i.e. SmoothGrad and Gradient Shap), while yielding the best correlations among the attribution methods. Hence, we choose IG-based score to assign the edge weights in NeurFlow.

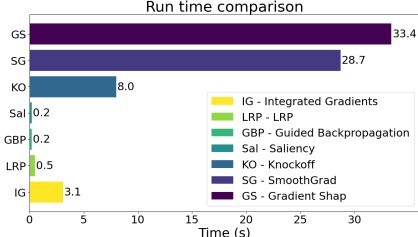

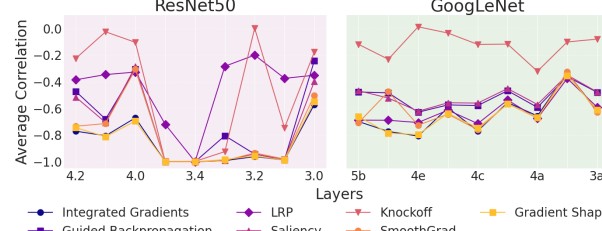

Figure 10: The comparison of average inference time across all layers in GoogLeNet on CPU.

Figure 11: The comparison of different attribution methods for edge weight quantification.

### D.2 NEURON GROUP RELATION WEIGHTS AGGREGATION

In this experiment, we compare our choice of summing the edge weights with averaging the edge weights in forming $W(G_i, G_j)$ in Section 3.5. Our aim is to verify that: *groups of neurons with higher sum of scores will have higher impact on a target neuron, regardless of the number of neurons in the group.*

We randomly sample 500 groups of neurons of varying sizes, ranging from $\{1, 5, 10, 20, 50\}$. For a target neuron in the upper layer, we analyzed the correlation between the loss function (defined in 4 and two metrics: the average edge weights within each group and our original scoring method, which sums the edge weights of neurons in the group. Higher absolute correlation values indicate a more effective scoring method. The results in figure 12 are the average of 10 neurons of different labels in both GoogLeNet and ResNet50.

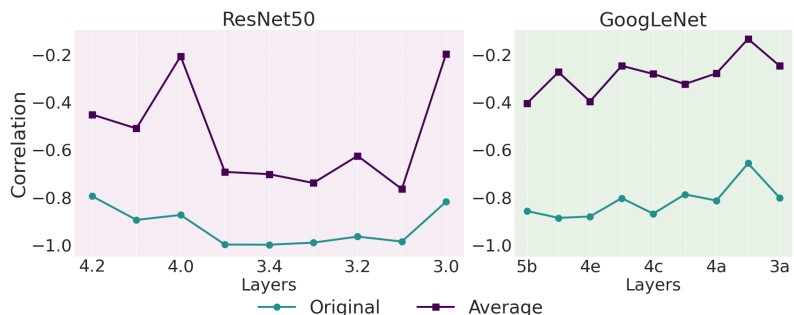

Figure 12: The correlations across 10 layers of our proposed aggregation (denoted as Original) and average aggregation (denoted as Average) on GoogLeNet and ResNet50.

### D.3 QUALITATIVE COMPARISON OF IMAGE DEBUGGING WITH NEUCEPT

We conduct a qualitative experiment to compare the set of critical neurons identified by Vu et al. (2022) (the core concept neuron w.r.t the output logit of the model) and our set in the image debugging experiment. Specifically, following the setups in the experiment in section 5.1, we identify the top $\tau = 16$ core concept neurons at layer 4.2 of ResNet50 for both methods, which are used to determine the top-2 groups of core concept neurons for a given misclassified image. Groups of neurons were identified following the methodology described in section 3.5, where the groups with the highest metric scores (defined in equation 5) are selected. Furthermore, to quantify the contributions of the selected groups to the model output, we mask all of neurons in each groups and measure the changes of probability of the final predictions. The higher the changes, the more "critical" the groups of neurons. We select three classes, without cherry-picking, namely: Bald Eagle, Great White Shark, and Bee (corresponding to the classes in figure 7, 8, and 9). The results are presented in figure 13, 14, and 15.

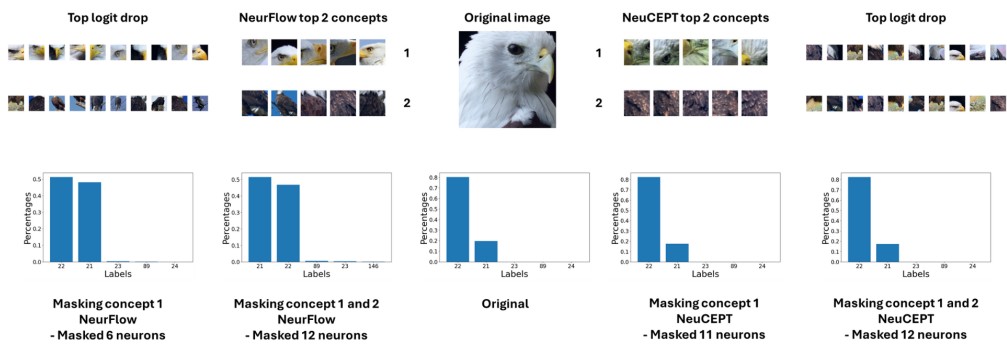

Figure 13: The comparison of the top-2 groups of neurons with the highest metric score of our method and Vu et al. (2022) on class *Bald eagle*. The top logit drop images of NeurFlow are more resemble the original concept (i.e. NeurFlow concept 1 vs NeuCEPT concept 1). And the prediction probability changes when masking our core concept neurons are more significant while masking fewer neurons.

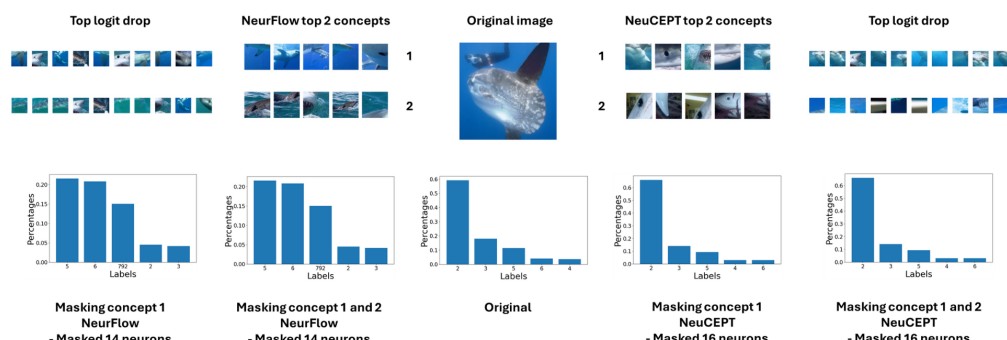

Figure 14: The comparison of the top-2 groups of neurons with the highest metric score of our method and Vu et al. (2022) on class *Great white shark*. The top logit drop images of NeurFlow are more resemble the original concept (i.e. NeurFlow concept 2 vs NeuCEPT concept 2). And the prediction probability changes when masking our core concept neurons are more significant while masking fewer neurons.

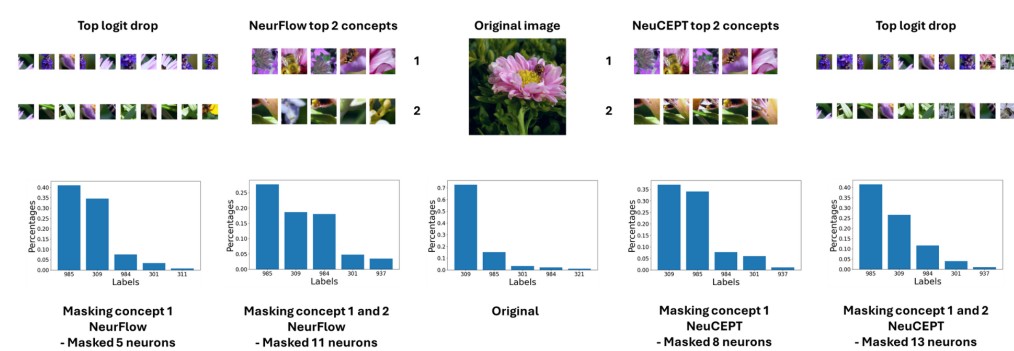

Figure 15: The comparison of the top-2 groups of neurons with the highest metric score of our method and Vu et al. (2022) on class *Bee*. The top logit drop images of both methods are similar to the exemplary image of the concept. And, both methods are able to alter the prediction of the model.

Qualitatively, we observed that our method identified the top-2 concepts more closely resembling the original images. Additionally, our top logit drop images (i.e., "images showing the largest decrease in the target logit value" as described in 5.1) better matched the representative examples of the identified concepts. Furthermore, masking the core concept neuron groups identified by our method resulted in more significant changes to the prediction probabilities, using fewer neurons, compared to the groups identified by NeuCEPT (Vu et al., 2022). For instance, with the labels Bald Eagle and Great White Shark, masking NeuCEPT's core concept neurons had no effect on prediction probabilities, whereas masking the neurons identified by our method substantially altered the predictions. These findings suggest that our approach identifies more impactful neurons and concepts directly related to the model's predictions compared to NeuCEPT.

## D.4 QUANTITATIVE COMPARISON OF CORE CONCEPT NEURONS OF THE MODEL OUTPUT

We run an experiment to further verify: although our method focuses on the set of core concept neurons w.r.t a specific target neuron, our identified neurons also have strong influence to the performance of the model.

Specifically, we evaluate the overlaps between our core concept neurons and the critical neurons (which are specifically designed to find important neurons for the model output) determined by Khakzar et al. (2021a) and Vu et al. (2022), then average the results across all layers of ResNet50 and GoogLeNet of 10 random classes. The numbers of core concept neurons are set to be the same for all three methods. We measure the $F_1$ scores of the overlaps, which are shown in table 3. The

Table 3: Overlapping ratio of critical neurons between NeuronMCT (Khakzar et al., 2021a), NeuCEPT (Vu et al., 2022), and core concept neurons of NeurFlow

| Overlap | NeuronMCT-NeurFlow | NeuronMCT-NeuCEPT | NeurFlow-NeuCEPT |
|---------|--------------------|--------------------|-------------------|
| ResNet50 | 0.72 | 0.48 | 0.49 |
| GoogLeNet | 0.79 | 0.55 | 0.56 |

Table 4: Average subtraction of the losses. Negative means our loss is better and vice versa

| Model | Average Subtraction of the Losses |
|-------|-----------------------------------|
| ResNet50 | -0.082 |
| GoogLeNet | -0.013 |

results imply that NeurFlow contains mostly similar core concept neurons to NeuronMCT while not directly identifying core concept neurons of the output.

### D.5    QUANTITATIVE COMPARISON OF CORE CONCEPT NEURONS OF A TARGET NEURON

We assess our method of identifying core concept neurons given a specific target neuron with the method used in Cammarata et al. (2020). In Cammarata et al. (2020), neurons are ranked based on the top neurons with the highest $L_2$ weights connected to the target neuron. Note that this method is not applicable in other experiments since calculating weight magnitude is limited to consecutive layers.

For this comparison, we identify the top $\tau = 16$ core concept neurons in two consecutive layers (separated by one convolution layer, as per the setup in Cammarata et al. (2020)) using both methods. We then knock out these core concept neurons to observe how the target neuron's concept is affected. The extent of this change is quantified by the loss function defined in 4, where a lower loss indicates better performance. We randomly selected 100 neurons across 10 different convolution layers from both models and calculated the average difference in losses between the two methods. A negative result indicates our method produces a better loss, while a positive result indicates otherwise.

The results are summarized in table 4. These findings demonstrate that our method is more effective at identifying core concept neurons. Additionally, gradient-based approaches are more versatile, as they can be applied to non-consecutive layers (e.g., ResNet Block 4.2 → ResNet Block 4.1 in our experiments), whereas the $L_2$-weight-based approach is limited to consecutive layers.

### D.6    DEPENDENCE ON THE CHOICES OF $\tau$

**The trade-off of the parameter $\tau$:** In this experiment, we aim to study the choices of parameter $\tau$ on the set of core concept neurons of a model. Specifically, in the experiment "Fidelity of core concept neurons", the choice of $\tau$ can be seen as a trade-off between simplicity (the number of core concept neurons) and performance (the accuracy of the prediction when retaining only the core concept neurons). However, for $\tau = 4, 8$ the results are vary across our tested models. We conduct additional experiment to highlight that for sufficiently large $\tau$, the results are less dependent on the parameter.

We evaluate on 10 different labels with the same setups as in the experiment "Fidelity of core concept neurons" for $\tau = 20, 24$. The results in figure 16 show that with these higher $\tau$ values, the performance drops of the model become negligible. Furthermore, the differences between retaining for $\tau = 20$ and $\tau = 24$ at all layers are minimal, suggesting that the dependence on $\tau$ decreases as we increase the value.

**Completeness of core concept neurons on the output:** Additionally, we run an experiment to assess the completeness of NeurFlow in identifying the important neurons for the model's output. By greedily adding 50% more neurons in each layer, of which the neurons are ranked by the importance scores defined in Khakzar et al. (2021a). The higher the scores, the stronger the influence on the

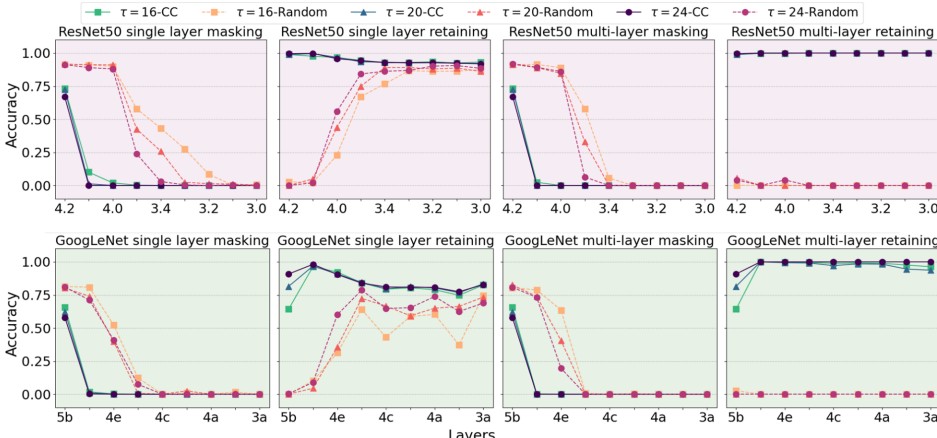

Figure 16: Effects of neuron groups on model's performance for $\tau = 16, 20, 24$. The effect of increasing $\tau$ are negligible for most of the layers in both models.

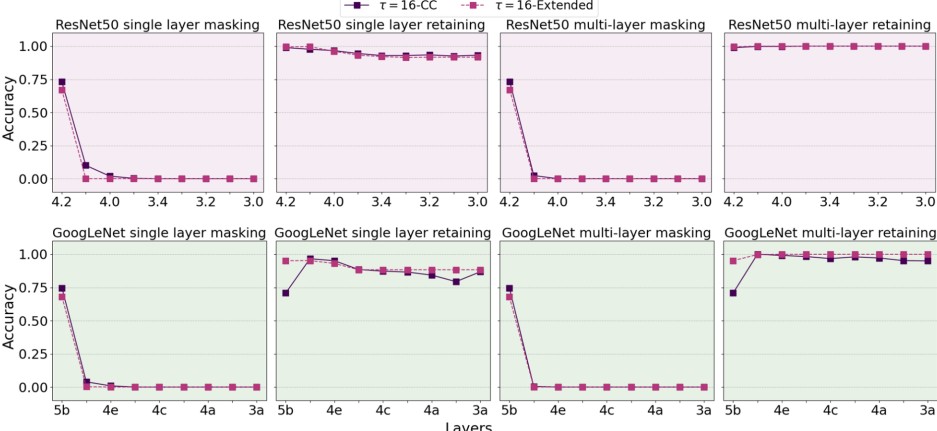

Figure 17: The comparison of the influences on models' performances of core concept neurons and the extended set of core concept neurons

prediction of the model. We then re-run the "Fidelity of core concept neurons" for $\tau = 16$ (denoted as "CC") and its extended version (50% more neurons - denoted as "Extended"). The results in figure 17 show that, for ResNet 50, adding non-core-concept neurons had almost no effect on improving model performance. For GoogLeNet, only in the most critical case (where the retaining operation is applied up to layer 5b), adding 50% more non-core-concept nodes led to an improvement in model performance by 25% only at layer 5b in the retaining setup. These results show that when $\tau$ is sufficiently large, our algorithm ensures completeness.

### D.7 DEPENDENCE ON THE CHOICES OF $k$

To evaluate the dependence of the results on the choice of $k$, we conducted additional experiments with various values of $k$ and measured the number of core concept neurons overlapping with the baseline setup of $k = 50$. Greater overlap indicates less dependence on the choice of $k$.

Table 5 summarizes the results with $\tau = 16$ (i.e., the maximum number of core concept neurons per target neuron is 16) and $k \in \{30, 40, 50, 60, 70, 90, 110, 130, 150, 170, 190\}$, evaluated across 50 random neurons. The results show that for all tested values of $k$, the overlap ratio is always at least $14/16$ ($> 86\%$), demonstrating that the results of our proposed algorithm are independent of the choice of $k$.

Table 5: The overlap of sets of core concept neurons of different $k$ compared to the baseline $k = 50$

| K | 30 | 40 | 50 | 60 | 70 | 90 | 110 | 130 | 150 | 170 | 190 |
|---|----|----|----|----|----|----|-----|-----|-----|-----|-----|
| GoogLeNet | 15.0 | 15.4 | **16.0** | 15.5 | 15.3 | 15.3 | 15.0 | 15.0 | 14.9 | 14.9 | 14.9 |
| ResNet50 | 14.9 | 15.6 | **16.0** | 15.6 | 15.3 | 14.7 | 14.5 | 14.3 | 14.1 | 14.0 | 14.0 |

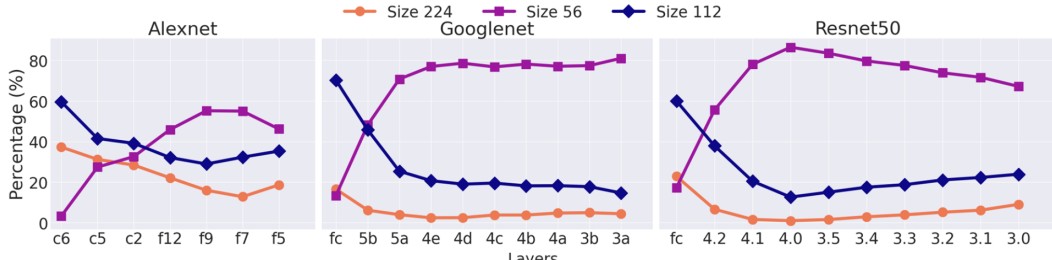

Figure 18: Illustration of the percentages of crop sizes in the concepts of core concept neurons. 50 random classes are assessed for three models and three different crop sizes. The layer names are abbreviated (e.g."feature.12" to "f12").

### D.8 Multiple crop sizes augmentation

For a target class $c$, our input dataset is created by randomly cropping the images that the DNN classified as class $c$, similar to Fel et al. (2023). However, since each neuron can detect feature at different granularity, we crop the images into multiple crop sizes in order to capture features at different levels. Intuitively, small crop sizes indicate low level while large crop sizes indicate high level features. In our experiments, we crop the original images into patches of three different sizes—100%, 50%, and 25% of the original dimensions. The cropping is performed using a sliding window with a 50% overlap, resulting in roughly 2500 patches in total.

Figure 18 shows the percentages of each crop size in the concepts of core concept neurons throughout the networks. As demonstrated, lower layer's neurons often activated on small crop size images and vice versa. This aligns with the common believe that high level features are detected at the later stages of DNNs. This approach can be improved further by including more complex augmentation methods. However, in this work, our main focus is functional of groups of neurons and their interactions.

## E Detailed algorithms

### E.1 Identifying core concept neurons and constructing neuron circuit

Algorithms 1, 2, and 3 provide detailed pseudocode for identifying core concept neurons, determining the semantic groups, and constructing the neuron circuit respectively.

---

**Algorithm 1** Identifying core concept neurons

**Input**: Target neuron $a$, dataset $\mathcal{D}$, constraint $\tau$
**Output**: Set of core concept neurons $S_a$

$\mathcal{V}_a \leftarrow \underset{\mathcal{V} \subset \mathcal{D}; |\mathcal{V}| = k}{\arg\max} \sum_{v \in \mathcal{V}} \phi_a(v)$

$T \leftarrow$ calculate $T(a, s_i, \mathcal{V}_a), \forall s_i \in \mathbb{S}$

$\mathbb{S}_a \leftarrow$ select the top-$\tau$ neurons with the highest $\|T\|$

**return** $\mathbb{S}_a$

---

---

**Algorithm 2** Determining semantic groups

---

**Input**: Neuron concept $\mathcal{V}_a$
**Parameter**: Max number of clusters $N_{cluster}$
**Output**: Semantic groups $\mathcal{V}_{a,j}, \ \forall j$

    $r(v_a^i) \leftarrow$ Calculate the representative vectors $\forall v_a^i \in \mathcal{V}_a$
    $best\_sil\_score \leftarrow -1$
    $best\_n \leftarrow$ Initialize
    **for** Number of clusters $n$ in $\{2, \ldots, N_{cluster}\}$ **do**
        $\mathcal{V}'_{a,j} \leftarrow$ Agglomerative clustering with $n$ clusters on $\{r(v_a^i), \ \forall v_a^i \in \mathcal{V}_a\}$
        $sil\_score \leftarrow$ calculate the Silhouettes score given the results of clustering
        **if** $best\_sil\_score < sil\_score$ **then**
            $best\_sil\_score \leftarrow sil\_score$
            $best\_n \leftarrow n$
        **end if**
    **end for**
    $\mathcal{V}_{a,j} \leftarrow$ Agglomerative clustering with $best\_n$ clusters on $\{r(v_a^i), \ \forall v_a^i \in \mathcal{V}_a\}$
    **return** $\mathcal{V}_{a,j}, \ \forall j \in \{1, \ldots, best\_n\}$

---

**Algorithm 3** Forming neuron circuit

---

**Input**: Logit neuron $a_c$, dataset $\mathcal{D}$, constraint $\tau$
**Output**: Neuron circuit $\mathcal{H}_c$

    $\mathcal{H}_c \leftarrow \{\,\}; S_L \leftarrow \{a_c\}; \mathcal{H}_c \leftarrow \mathcal{H}_c \cup S_L$
    **for** Layer $l$ in $\{L-1, \ldots, 2, 1\}$ **do**
        $S_l \leftarrow \{\,\}$
        **for** Target neuron $a$ in $S_{l+1}$ **do**
            $S_l \leftarrow S_l \cup$ Identify core concept neurons (Alg.1) of $a$
            $\mathcal{V}_{a,j} \leftarrow$ Determine semantic groups (Alg.2), $\forall j$
            $w(s_i, \mathcal{V}_{a,j}) \leftarrow T(a, s_i, \mathcal{V}_{a,j})/\sum_{s \in \mathbb{S}_a} \|T(a, s, \mathcal{V}_{a,j})\|$
        **end for**
        $\mathcal{H}_c \leftarrow \mathcal{H}_c \cup S_l$
    **end for**
    **return** $\mathcal{H}_c$

---

## F   IMAGE DEBUGGING SETUP

For an arbitrary input $v \in \mathcal{D}$, we want to see which parts of $v$ are detected by the group of neurons $G$. Thus, we crop the image into multiple crops, similar to what we do in Section 3.3. The crops, denoted as $v_i$ are passed into the model to get the activations, which we can then measure the metric $M(v_i, \mathbb{S}_G, \mathcal{D}), \ \forall v_i$. Then we can set a threshold for each group, so that, the crops with the scores above the threshold can be visualized.

However, since the metric can be greatly affected by only one neuron in the group (i.e one neuron with low activation leads to a low metric score), the metric is prone to outliers. Thus, we only assess the metric on the subset $\mathbb{S}'_G \subseteq \mathbb{S}_G$. In practice, $\mathbb{S}'_G$ contains the top-5 neurons that are closest to the group's center, where each neuron is represented as $\overrightarrow{r_{s_i,j}}$ for a neuron $s_i \in \mathbb{S}_G$ with the semantic group's index $j$. The center of the cluster is the average of all representative vectors, and the distance between a pair of neurons is evaluated using $l_2$ distance.

## G   MLLM PROMPT FOR AUTOMATIC CONCEPT LABELLING

In this section, we provide our prompts for reproducibility. We employ two types of prompt, which are responsible either captioning the common concepts in the exemplary images of a neuron concept, or describing how a NGC formed from NGCs at the preceding layers. Our prompts include three parts. Firstly, we provide a role for MLLM model, marked as *role description*. Secondly, the *main prompt* is presented where it shows the general instruction for the task that MLLM should do. The

*role description* and *main prompt* is the same for all setups. The last part is the *answer form* where we give specific instruction on how to generate appropriate captions and the template of the answer. The structure of the whole prompts are: *Role description + Main prompt + Answer form*.

## G.1 ROLE DESCRIPTION AND MAIN PROMPT

**Role descriptions:** "Act as an Image Captioning Language Model."

**Main prompt:**

"# Core Responsibilities:

- Analyze a set of similar images to identify common features.

- Generate descriptive captions that highlight these common features.

- You must adapt to detect both simple and complex features.

# Important notes:

- You don't have to generate captions for every image, focus on the common features.

- Outliers exist in the images, you could ignore them if they are not relevant to the common theme.

- You should describe the images with objective visual features, not subjective (like powerful or beautiful or scary etc., because these are only your opinion).

- You should only describe visual features, not the context or the story behind the images.

- You should keep a succinct caption, keep it one or two sentences long, that only describe a few most common features.

# Role Summary:

Your role is to provide accurate and coherent captions for a set of similar images by identifying and describing common features. These features can range from simple elements like edges and colors to complex patterns such as a specific object in a particular setting."

## G.2 ANSWER FORM

**Answer form for single concept captioning:**

*"# Answer form:*

- Common features: a list of features

- Caption: your caption in one or two sentences"

**Answer form for describing NGC's formation:**

*# Key note of the input:*

- There are many different groups of images, make sure you get the number of groups right.

- Each group of images has a common feature.

- The higher level feature is the first group.

- Other groups are lower level features that combine to form the higher level feature of the first group.

*# Key note of the output:*

- You should not only focus on the common features of the images but also describe how the features from the lower level groups combine to form the higher-level feature of the first group.

- You should focus on the common features that shared among both the high and low level.

*"# Step by step:*

- Find the lists of common features in Group 2, . . . , N.

- For each feature from those lists: match it with the features in Group 1.

- Some of the features in the lists might have no matches: they might be combined with others to form new features, match the features in Group 1 with some simple combination of the features in Group 2, ..., N (e.g. blue and green → blue-green, multiple curve orientations → a circle, two edges with different orientations → an angle, etc.).

- If you don't find any visual features that match, please don't describe features that is not presented, instead, you can say "There is no matches".

- From the matched features, derive the common features in Group 1.

- Generate caption for Group 1.

*# Answer form:*

- Group 1 Common Features: list of common features

- Group 2 Common Features: list of common features

- ...

- Group N Common Features: list of common features

Feature Evolution:

- Group 2: has feature A - match feature A in Group 1 (for Group 2 to N, if there is no matches, please say "There is no matches")

- ...

- Group N: has feature B - match feature B in Group 1

Caption: one or two sentences capturing the common features and their evolution"

