# OpenReview forum: "NeurFlow: Interpreting Neural Networks through Neuron Groups and Functional Interactions"
_ICLR.cc/2025/Conference — ICLR 2025 Poster_

### Official Review · Reviewer_KddZ · 2024-10-30

**Soundness:** 3
**Presentation:** 3
**Contribution:** 2
**Rating:** 6
**Confidence:** 4

**Summary:**

The paper proposes a novel framework, NeurFlow, to enhance the interpretability of deep neural networks (DNNs) by focusing on critical neuron groups and their interactions, while existing works focus more on interpreting individual neurons. The framework proposes constructing hierarchical circuits encapsulating critical neuron interactions across layers, providing a more holistic view of model decision-making.

The proposed method can be summarized in the following steps.
(1) Identifies critical neurons
(2) Clusters these neurons into groups
(3) Investigate the functions and interactions of these groups.

**Strengths:**

• The paper introduces a novel way to identify semantic groups by leveraging clustering. By clustering neuron responses into semantic groups, the proposed method can interpret the target model with more specified concepts.

• Interesting applications of the proposed method with easily understandable figures.

•  The paper is well-written and easy to understand, except for a few parts.

**Weaknesses:**

Although the paper showed novelty and interesting results, I still have some concerns about this paper.

Major remarks

1. Existing works like [1*] and [2*] investigate neuron groups as fundamental units for explaining the internal workings of deep neural networks. Especially, [1*] proposes a similar idea of identifying critical neurons from different layers with importance scores derived from integrated gradients, which should be compared. Also, what are the major strengths of this paper regarding [1*] and [2*]?

2. This paper aims to enhance the interpretability of deep neural networks (DNNs). However, the proposed method only evaluated on CNNs (ResNet and GoogLeNet). Does the proposed method also work in other architectures like ViTs?

Minor remarks

1. How many random crop patches are needed to guarantee the stable performance of NeurFlow?

2. Various ways of determining the importance score should be evaluated (i.e., various approximations of Shapley value)

3. Existing works like [3*] identify neuron responses with highly activating crop and original images. It also investigates basic groups of neurons for decision-making, which should be referred to in the paper.

4. In Figure 6, why do some layers(i.e., 2,3,4 in resnet and 3b, 4b, 4d in googlenet) have low correlations compared to the previous and subsequent layers? Can you share some insights regarding this result?

5. Shouldn't it be "knocking out these neurons" In line 181?

References

[1*] Khakzar, Ashkan, et al. "Neural response interpretation through the lens of critical pathways." CVPR 2021.

[2*] Achtibat, Reduan, et al. "From attribution maps to human-understandable explanations through concept relevance propagation." Nature Machine Intelligence 5.9 (2023)

[3*] Kalibhat, Neha, et al. "Identifying interpretable subspaces in image representations." ICML 2023.

**Questions:**

Please address or explain the remarks in the above Paper's Weaknesses section.

---

> ### Author Response · Authors · 2024-11-21
> **Response to Reviewer KddZ**
>
> &nbsp;
>
> **We sincerely thank the Reviewer for the very detailed and thoughtful questions.
> Below are our responses.**
>
> >**Q1. Existing works [1, 2] investigate neuron groups as fundamental units for explaining the internal workings of deep neural networks. Especially, [1] proposes a similar idea of identifying critical neurons from different layers with importance scores derived from integrated gradients, which should be compared. What are the major strengths of this paper regarding [1] and [2]?**
>
> **Response:** [1] proposes a method to identify the **critical pathway** (a sub-network) that encodes essential information from an input. Meanwhile, [2] introduces a method for constructing **Concept Composition Graphs,** which decompose a concept of interest into lower-layer concepts.
> In short, both methods focus on **interpreting the model’s response (given a specific input)** rather than identifying the relationships between neurons.
> Specifically, [1] prunes the unimportant neurons (based on neurons’ contributions to the response) to form a sparse sub-network, while [2] labels each neuron with a human-understandable concept, then performs attribution (using LRP) with a condition of attributing only to neurons that contain the target concept, to build a concept graph that illustrates how a high-level concept is composed of lower-level concepts.
>
> In contrast, **our approach** emphasizes **the relationships between groups of neurons** and provides a systematic solution for constructing circuits of neuron groups, which explain **the internal interactions between layers.**
>
> In addition, compared with our approach, [2] suffers from the need for human annotation and manual operation. In contrast, our method is fully automated and does not rely on predefined concepts.
> Moreover, it is worth noting that [2] overlooks the polysemantic nature of neurons, assigning only one concept per neuron. By contrast, our method is the only one that uses neuron groups as the fundamental units to explain the internal workings of deep neural networks, thereby addressing the phenomenon of polysemanticity effectively.
>
> &nbsp;
>
> >**Q2. This paper aims to enhance the interpretability of (DNNs), but only evaluated on CNNs. Does the proposed method also work in other architectures like ViTs?**
>
> **Response:** Our research primarily focuses on CNNs, a type of network widely used in many state-of-the-art models in this domain  [4, 5, 6, 7].  We will revise the final draft of the paper to clearly specify the scope of the study.
>
> The framework we propose can also be applied to Transformer models. We will include discussions on this topic in the Appendix.

---

> ### Author Response · Authors · 2024-11-21
> **Response to Reviewer KddZ (con't)**
>
> &nbsp;
>
> **Minor remarks:**
>
> >**M1. How many random crop patches are needed to guarantee the stable performance of NeurFlow?**
>
> **Response:** While it is difficult to determine the exact minimum number, our experiment requires only $50$ original images per class (the standard number in the validation set). These images are cropped into patches of three different sizes—$100\\%$, $50\\%$, and $25\\%$ of the original dimensions. The cropping is performed using a sliding window with a $50\\%$ overlap, resulting in approximately $2500$ patches in total.
>
> &nbsp;
>
> >**M2. Various ways of determining the importance score should be evaluated (i.e., various approximations of Shapley value)**
>
> **Response:** We have compared our method with other gradient-based attribution techniques in Appendix A.1 (referenced in Lines 430-431). Specifically, we evaluate four widely used pixel attribution methods: LRP [8], Guided Backpropagation [9], SmoothGrad [10], and Saliency [11]. Additionally, we assess the attribution method used in [12]. The results presented in Figures 10 and 11 indicate that our score determination method has a smaller runtime compared to the follow-up method (i.e., SmoothGrad), while providing the highest correlations among the attribution techniques.
>
> Now, we conducted an additional experiment comparing our score determination method with Gradient Shap [13]. The results show that while Gradient Shap achieves similar performance to Integrated Gradient (as seen in [Link](https://anonymous.4open.science/r/NeurFlow-Rebuttal-images-3F6E/Adding%20Gradient%20Shap%20for%20correlation.png)), its inference time is significantly higher than that of our proposed method ([Link](https://anonymous.4open.science/r/NeurFlow-Rebuttal-images-3F6E/Adding%20inference%20time%20Gradient%20Shap.png)).
>
> &nbsp;
>
> >**M3. Existing works like [3] identify neuron responses with highly activating crop and original images. It also investigates basic groups of neurons for decision-making, which should be referred to in the paper.**
>
> **Response:** We thank the reviewer for the very thoughtful comment. We will cite the mentioned paper in the final manuscript.
>
> &nbsp;
>
> >**M4. Shouldn't it be "knocking out these neurons" In line 181?**
>
> **Response:** We thank the Reviewer for pointing out this typo. We will fix it in the final version.
>
> &nbsp;
>
> **If our explanations have addressed the Reviewer's questions, we kindly hope the Reviewer might consider increasing the score.**
>
> **Should there be any remaining issues or points of concern, please let us know. We will gladly follow up to provide further clarification.**
>
> &nbsp;
>
> **References:**
>
> [1] Khakzar, Ashkan, et al. "Neural response interpretation through the lens of critical pathways." CVPR 2021.
>
> [2] Achtibat, Reduan, et al. "From attribution maps to human-understandable explanations through concept relevance propagation." Nature Machine Intelligence 5.9 (2023).
>
> [3] Kalibhat, Neha, et al. "Identifying interpretable subspaces in image representations." ICML 2023.
>
> [4] Nguyen, Anh, et al. "Multifaceted feature visualization: Uncovering the different types of features learned by each neuron in deep neural networks." arXiv 2016.
>
> [5] ChrisOlah, et al. Zoom in:An introduction to circuits. Distill 2020.
>
> [6] Jesse Mu et al. Compositional explanations of neurons. NeurIPS 2020.
>
> [7] Laura O’Mahony, et al. Disentangling neuron representations with concept vectors.  CVPR 2023.
>
> [8] Bach, Sebastian, et al. On pixel-wise explanations for non-linear classifier decisions by layer-wise relevance propagation. PloS one 2015.
>
> [9] Springenberg, Jost Tobias, et al. Striving for simplicity: The all convolutional net. arXiv 2014.
>
> [10] Smilkov, Daniel, et al. Smoothgrad: removing noise by adding noise. arXiv 2017.
>
> [11] Simonyan, Karen. Deep inside convolutional networks: Visualising image classification models and saliency maps. arXiv 2013.
>
> [12] Vu, Minh N., NeuCEPT: Locally Discover Neural Networks' Mechanism via Critical Neurons Identification with Precision Guarantee. arXiv 2022.
>
> [13] Lundberg, Scott. "A unified approach to interpreting model predictions." arXiv preprint arXiv:1705.07874 (2017).

---

> ### Author Response · Authors · 2024-11-22
> **Follow-up**
>
> Dear Reviewer **KddZ,**
>
> We hope our response has thoroughly addressed your questions.
>
> We would greatly appreciate any further feedback or additional questions you may have.
>
> Please let us know.
>
> Thank you once again for your thoughtful insights and consideration.
>
> Best regards,
>
> Authors

---

> > ### Comment · Reviewer_KddZ · 2024-11-25
> >
> > I want to thank the authors for their detailed responses.
> >
> > Unfortunately, my concern in Q1 is still unclear. Intrinsically, subnetwork and graphs are also ways to provide(or to organize) the relationships between groups of neurons. Also, [2] explains the internal interactions between layers both locally(i.e., per sample) and globally(i.e., per model). The fundamental difference between NeurFlow and [1,2] still needs to be clarified. If the NeurFlow approach is similar to [1,2], further experiments to show the advantages of the NeurFlow would be helpful.

---

> ### Author Response · Authors · 2024-11-25
> **Follow-up**
>
> **We would like to thank the Reviewer for taking the time to discuss with us.
> We are pleased to respond to your questions as follows.**
>
> &nbsp;
>
> >**F1. Unfortunately, my concern in Q1 is still unclear. Intrinsically, subnetwork and graphs are also ways to provide(or to organize) the relationships between groups of neurons. Also, [2] explains the internal interactions between layers both locally(i.e., per sample) and globally(i.e., per model). The fundamental difference between NeurFlow and [1,2] still needs to be clarified. If the NeurFlow approach is similar to [1,2], further experiments to show the advantages of the NeurFlow would be helpful.**
>
> **Response:** We would like to emphasize that the key distinction between NeurFlow and [1], [2] is that we seek to ultimately **construct the concept circuit**, which is defined in Section 3.5, in the last paragraph. Intuitively, this circuit consists of groups of neurons, where each group corresponds to an interpretable concept. Since many neurons can correspond to the same concept [3], we create a circuit from semantic clusters of neurons for a more compact and informative explanation. This step is not present in existing works (such as [1,2]), which instead consider graphs of individual neurons. We further elaborate the key differences as follows:
>
> **Regarding [1]:** their approach involves keeping the neurons with the highest scores relative to the output, with the remaining graph labeled as *"critical pathways."* As a result, their method **only measures the relationship between a neuron and the output**, and there are **no quantifications between an intermediate neuron and the neurons in the layer below**.
>
> We acknowledged your concern: *“Intrinsically, subnetwork and graphs are also ways to provide (or to organize) the relationships between groups of neurons”.* However, the **connections between the neurons** in the graph (or "critical pathway") of the retained neurons are **actually the weights of the original model,** which do not directly reflect the relationships between the neurons.
>
>
>  &nbsp;
>
> **Regarding [2]:** Due to the difference in objectives (while we build the concept circuits and [2] aims to find critical neurons directly to the target), their approach for constructing the graph and interpreting its meaning is entirely different from ours. In addition, it also has a significant drawback, as we mentioned in our previous response: they must manually assign a human-understandable concept to each feature map. This step is essential because the defined score (conditional LRP [2]) is only attributed to feature maps associated with a specific concept.
>
> We further elaborate the difference between the two approaches below.
>
> **Pipeline of [2]:**
>
> * First, they are provided with a set of low-level concepts (although they did not explain how to define these concepts).
> * Then, for each feature map, they use ActMax or RelMax to identify the images with the highest activation (ActMax) or the most relevance (RelMax) to the feature map. Based on this set of images, they assign the meaning to the feature map as the concept that appears most frequently within the image set.
> * For a target class (e.g., dog), they choose the predefined low-level concepts associated with the class (e.g., eyes, etc.). Then, they apply LRP to attribute values only to the feature maps containing the relevant concepts for the class. This produces a graph of concepts, with quantification between the low-level concepts.
>
> **Pipeline of ours:**
>
> * First, we randomly crop the data from the class of interest into patches, which we refer to as visual features. It’s important to note that these visual features are not concepts and are distinct from the low-level concepts used in [2].
> * We use the visual features to iteratively identify critical neurons and compute the score between critical neurons and the target neurons based on the Integrated Gradient score. This process results in a hypertree, as outlined in the paper.
> * We cluster the visual features into semantic groups. MLLM is then used to label the concepts associated with each semantic group.
>
> In summary, the key differences are as follows: In [2], the concepts are predefined and used to construct the graph, while in our approach, the graph is formed based on the relationships between neurons, and concepts are defined only after the graph is built.
>
> **The pipelines of the two methods are essentially reversed with two different objectives.** Additionally, it is important to highlight that our method is fully automated and does not depend on predefined concepts, whereas [2] relies on predefined, labeled low-level concepts.
>
> &nbsp;
>
> **We sincerely hope that our responses have addressed your concerns.**
>
> **If there is anything further you would like to discuss or clarify, please let us know.
> We would be delighted to continue the discussion.**

---

> > ### Comment · Reviewer_KddZ · 2024-11-26
> >
> > Thank you for your detailed responses. Some of my concerns have been addressed. I will raise my scores.

---

> > > ### Author Response · Authors · 2024-11-26
> > > **Thank you for valuable comments and for improving our scores**
> > >
> > > We sincerely appreciate the Reviewer's time and effort in engaging with us and providing valuable, constructive feedback.
> > >
> > > We are grateful to the Reviewer for improving our scores.
> > >
> > > All the discussed content will be incorporated into the revised manuscript.
> > >
> > > Best regards,
> > >
> > > Authors

---

### Official Review · Reviewer_qMvj · 2024-11-02

**Soundness:** 2
**Presentation:** 3
**Contribution:** 4
**Rating:** 6
**Confidence:** 4

**Summary:**

The paper proposes NeuroFlow, a framework designed to identify the set of connected neurons (i.e., circuit) that most strongly influence the predictions of a target class. The framework begins by clustering neurons based on their semantics and then reconstructs the circuit through interactions between clusters across the network structure. The framework's main steps are as follows:

1. For a given neuron at layer L, the critical neurons for its output at layer L-1 are identified. These critical neurons are the ones whose removal causes the highest change in the top-k patches associated with the target neuron. This set is approximated by an algorithm that sums attributions (calculated using Integrated Gradients) for the identified top-k patches.
2. The Top-k patches activated by the neuron are clustered into semantic groups using agglomerative clustering, where each cluster represents a semantic concept captured by the neuron.
3. The algorithm calculates the contribution of each critical neuron in the preceding layer to each semantic group identified in the previous step.
4. Semantic groups from the entire layer are clustered based on their similarity, and neurons that share at least one clustered semantic group are clustered together.
5. A multimodal LLM labels the clustered semantic groups
6. The concept circuit is generated using the connections and the attributions of the critical neurons within each semantic group.

The proposed framework is evaluated by assessing (1) the approximation accuracy of the heuristic used to compute critical neurons, (2) the impact of critical neurons with respect to a random baseline, and (3) the correlation between attribution scores and accuracy loss when neurons are removed. The paper also demonstrates an application of NeuroFlow to the problem of bias detection.

**Strengths:**

- The paper is well structured, clear, and quite easy to follow (with some caveats noted in the weaknesses section)
- The polysemantic nature of neurons is often overlooked in the current literature, particularly in works focused on circuits. The paper's attention to this topic, even if only partially addressed (see weaknesses), is highly relevant and of interest to the ICLR community.
- Some of the characteristics of the frameworks are both novel and promising. In particular, step (3), the possibility for a neuron to be “active” in multiple clusters, and the connections between semantic groups are a clever way to deal with polysemantic neurons.
- The integration of previously explored directions into a unified framework offers a fresh perspective on the topic.

**Weaknesses:**

The main weakness of the paper regards the evaluation and supporting experiments. There is a mismatch between the strong claims and the evidence provided. The current evaluation adequately validates the framework’s design choices. However, it does not place the work in the broader context of the ongoing research on this topic. Given substantial improvements in this direction, I would be willing to increase the score.

In more detail, several steps of the framework have been explored in prior literature, albeit sometimes with different objectives. Analyzing the connections between the outcomes of these steps and findings from other approaches would greatly strengthen the paper. Below are some specific examples of this point.
- Step 1 identifies critical neurons. The way in which they are identified (in relation to the top-k patches) seems novel to me. However, the concept of critical neurons for a particular prediction or a class has already been studied in the literature. For example, authors of [1]  identify class-specific critical neurons. They found that a small subset of neurons, when removed, leads to significant shifts in prediction. Similarly,  (Vu et al., 2022) explore related concepts.  How do the findings in these papers relate to those in this work? Is there an overlap between their critical neurons and those identified here? If not, what could explain the differences?
- The paper claims to “demonstrate that for a particular task, only a subset of neurons—referred to as critical neurons—have a substantial influence on the model’s performance”.  However, the experiments only show that, on average (but not always), the identified critical neurons have a greater impact than randomly selected neurons. This mismatch should be addressed either by providing additional experiments that prove that no other neurons outside the critical impact the class or by lowering the claim.
- A large body of literature explores the extraction of circuits (like the ones cited in the paper). Although the circuits’ structures may differ, it would be valuable to compare their components. Do they share the same neurons? What important features are missing in prior approaches that the proposed framework captures? This would help clarify the unique contributions of this work.
- The point raised by the authors about “groups of neurons within each layer also collectively encode the same concept” is quite well known. On this topic, a highly related work is [2], which identifies collaborative neurons responsible for the recognition of a concept. Similarly to the previous points, it would be great to see a comparison in terms of finding between this paper and the cited related work.


Other weaknesses that didn’t impact my current score:
- Polysemantic behavior is not limited to the highest activations. Several previous works indicate that neurons can recognize concepts at the lowest activations or across a range of activations [3,4,5]. However, the current framework relies upon and analyzes only the highest activations to compute the “visual features”. Therefore it is not guaranteed that the critical neurons extracted are the only ones contributing to the recognition of those concepts. Note that some alternative approaches [1,  (Vu et al., 2022)] for computing critical neurons are immune to this problem. Given the emphasis on polysemantic behavior, I encourage the authors to acknowledge this aspect as a limitation, a future direction, or to provide a more generalized formulation of the framework applicable to a broader range of settings.
- Some terminology in Section 3.2 could be confusing for practitioners unfamiliar with the domain. Specifically, terms like "neuron concept," "top-kkk patches visual features," and "top-kkk patches" may cause confusion. For example, definition 1 defines the set of top-k patches as the “neuron concept”. That name implies that these patches represent a single concept, whereas, in later sections, they denote a set of visual features (or concepts) recognized by the neuron. Based on the equation and the paper's narrative,"top-k patches" would be a better term here.  The words “visual features” and patches are used interchangeably throughout the text, even if they are not exactly the same. Traditionally, “visual features” refer to the semantic: the same visual feature can be captured by several patches, so there could be five patches that identify one visual feature. For clarity, I suggest using "patches" where possible and reserving "visual features" for references to their semantic meanings. For instance, lines 201-205 would be clearer if the first and last occurrences of "visual features" were replaced with "patches."

[1] Amirata Ghorbani and James Zou. 2020. Neuron shapley: discovering the responsible neurons. In Proceedings of the 34th International Conference on Neural Information Processing Systems (NIPS '20). Curran Associates Inc., Red Hook, NY, USA, Article 497, 5922–5932.

[2] Wang, A., Lee, W., & Qi, X. (2022). HINT: Hierarchical Neuron Concept Explainer. 2022 IEEE/CVF Conference on Computer Vision and Pattern Recognition (CVPR), 10244-10254.

[3] Biagio La Rosa, Leilani H. Gilpin, and Roberto Capobianco. 2024. Towards a fuller understanding of neurons with clustered compositional explanations. In Proceedings of the 37th International Conference on Neural Information Processing Systems (NIPS '23). Curran Associates Inc., Red Hook, NY, USA, Article 3082, 70333–70354.

[4] https://transformer-circuits.pub/2023/monosemantic-features/index.html

[5] Tuomas Oikarinen and Tsui-Wei Weng. Linear Explanations for Individual Neurons. ICML 2024

**Questions:**

Please check the weaknesses section. Several parts include direct or indirect questions for the authors and I would appreciate their answer and clarification.

---

> ### Author Response · Authors · 2024-11-21
> **Response to Reviewer qMvj (part 1/3)**
>
> &nbsp;
>
> **We sincerely thank the Reviewer for highlighting relevant works and providing valuable feedback. Below, we address the distinctions between our approach and existing literature to clarify our novelty and contributions.**
>
> >**Q1. Step 1 identifies critical neurons. The way in which they are identified (in relation to the top-k patches) seems novel to me. However, the concept of critical neurons for a particular prediction or a class has already been studied in the literature. For example, authors of [1] identify class-specific critical neurons. They found that a small subset of neurons, when removed, leads to significant shifts in prediction. Similarly, (Vu et al., 2022) explore related concepts. How do the findings in these papers relate to those in this work? Is there an overlap between their critical neurons and those identified here? If not, what could explain the differences?**
>
>
> **Response:** The definition of *“critical neurons”* in our work is indeed different from those defined by literature. To be specific, **existing works** focus more on the relationship between neurons and the final results, and define the “critical neurons” as those **critical to the model’s final prediction results.**
>
>
> In contrast, **ours** focuses on inter-layer relationships.
> As such, we define critical neurons as those **critical to a specific neuron in the following layer**.
> By this way, our framework enables an understanding of the internal workings of neural networks and can be utilized for various applications (as detailed in Section 5), such as image debugging and the automated detection of layer-by-layer relationships.
>
>
> &nbsp;
>
> >**Q2. Mismatch between the strong claims and the evidence provided
> The paper claims to “demonstrate that for a particular task, only a subset of neurons—referred to as critical neurons—have a substantial influence on the model’s performance”. The experiments only show that the identified critical neurons have a greater impact than randomly selected neurons. This mismatch should be addressed either by providing additional experiments that prove that no other neurons outside the critical impact the class or by lowering the claim.**
>
>
>
> **Response:** We thank the Reviewer for the comment.
>
>
> Firstly, since our work focuses on identifying the critical neurons for a target neuron (rather than those critical to the output), we will revise the above statement to precisely reflect our definition of critical neurons and their inter-layer relationships.
>
>
> However, because the set of all critical neurons ultimately influences the model's final prediction, we conducted an additional experiment to demonstrate that “including more non-critical neurons into the set of critical neurons identified by our algorithm does not significantly improve the model's performance.”
>
> Specifically, we performed the “Fidelity of critical neurons” experiment with $\tau = 16$, where we added $50\\%$ more non-critical nodes, and assessed the impact on model accuracy.
> The non-critical neurons were selected greedily, using the nodes with the highest scores ranked by NeuronMCT (as suggested by Reviewer 4).
> The results are provided at [Link](https://anonymous.4open.science/r/NeurFlow-Rebuttal-images-3F6E/Adding%20more%20neurons%20for%20figure%205.png).
>
> The results show that, for ResNet 50, adding non-critical neurons had almost no effect on improving model performance.
> For GoogleNet, only in the most critical case (where the retaining operation is applied up to layer 5b), adding $50\\%$ more non-critical nodes led to an improvement in model performance by $25\\%$ only at layer 5b in the retaining setup. These results show that when $\tau$ is sufficiently large, our algorithm ensures completeness.

---

> ### Author Response · Authors · 2024-11-21
> **Response to Reviewer qMvj (part 2/3)**
>
> >**Q3. A large body of literature explores the extraction of circuits (like the ones cited in the paper). Although the circuits’ structures may differ, it would be valuable to compare their components. Do they share the same neurons? What important features are missing in prior approaches that the proposed framework captures? This would help clarify the unique contributions of this work.**
>
> **Response:** We thank the Reviewer for a constructive comment.
> The key differences between our approach and existing works are as follows:
> * To our knowledge, we are the first to systematically identify circuits of neurons and their inter-layer relationships.
> * Our approach is the only one that employs neuron groups as the fundamental units for explaining the internal workings of deep neural networks, thereby addressing the polysemantic phenomenon.
> * Our approach focuses solely on critical neurons instead of performing a brute-force search across all neurons (Cammarata et al. (2020) [7]). This significantly reduces computational costs while ensuring the scalability of the solution.
>
> To clarify the distinctions between our work and other neuron- and circuit-based studies cited in the paper (Lines 092-098), we provide a summary of their main objectives below:
> 1. Nguyen et al. (2016) [3], O’Mahony et al. (2023) [4], Mu & Andreas (2020) [5]: These works focus on labeling the semantics of individual neurons but do not involve constructing neuron circuits.
> 2. Bykov et al. (2024) [6]: This method primarily aims to learn compositional concepts that explain neuron representations rather than forming circuits. While the authors demonstrated a way to manually construct a circuit, their circuit represents concepts rather than neurons, focusing on how compositional concepts emerge from atomic ones.
> 3. Cammarata et al. (2020) [7]: This approach involves labeling neurons individually and manually connecting them to form circuits, which results in significant computational overhead.  Our circuit is different from this approach as we focus on the circuit of groups of neurons, rather than individual neurons.
>
> &nbsp;
>
> >**Q4. The point raised by the authors about “groups of neurons within each layer also collectively encode the same concept” is quite well known. On this topic, a highly related work is [2], which identifies collaborative neurons responsible for the recognition of a concept. Similarly to the previous points, it would be great to see a comparison in terms of finding between this paper and the cited related work.**
>
> **Response:** The key distinction between [2] and our approach lies in their focus.
>
> [2] emphasizes identifying relationships between neurons and concepts. As stated in [2], “HINT identifies collaborative neurons responsible for one concept and multimodal neurons pertinent to different concepts”.
> However, [2] does not address the relationships between neurons themselves, meaning it cannot quantify these connections or construct a neuron circuit.
>
> In contrast, our approach centers on inter-layer relationships between neuron groups. We create circuits of neuron groups to explain how critical neuron groups function and interact to accomplish a specific task.
>
> For instance, our framework enables debugging false prediction issues (as demonstrated in Section 5.1), a capability that is not achievable by using [2].

---

> ### Author Response · Authors · 2024-11-21
> **Response to Reviewer qMvj (part 3/3)**
>
> &nbsp;
>
> **Other issues that didn’t impact the Reviewer’s current score:**
>
> >**M1. Polysemantic behavior is not limited to the highest activations. Given the emphasis on polysemantic behavior, I encourage the authors to acknowledge this aspect as a limitation, a future direction, or to provide a more generalized formulation of the framework applicable to a broader range of settings.**
>
> **Response:** We thank the Reviewer for the constructive comment.
> Our framework includes several hyperparameters to enhance flexibility, such as $\tau$ (the maximum number of critical neurons per target neuron).
> We will investigate the issue raised by the Reviewer and improve the framework to enhance its robustness.
> Additionally, we will include a discussion on this matter in the final manuscript.
>
> &nbsp;
>
> >**M2. Some terminology in Section 3.2 could be confusing for practitioners unfamiliar with the domain. Specifically, terms like "neuron concept," "top-kkk patches visual features," and "top-kkk patches" may cause confusion. For clarity, I suggest using "patches" where possible and reserving "visual features" for references to their semantic meanings.**
>
> **Response:** We sincerely thank the Reviewer for such a thoughtful comment.
> We will revise the wording appropriately (as suggested by the Reviewer) to avoid any potential misunderstanding for the readers.
>
> &nbsp;
>
>
> **If our explanations have addressed the Reviewer's questions, we kindly hope the Reviewer might consider increasing our score.**
>
> **Should there be any remaining issues or points of concern, please let us know. We will gladly follow up to provide further clarification.**
>
> &nbsp;
>
> **References:**
>
> [1] Amirata Ghorbani et al. Neuron shapley: discovering the responsible neurons. NeurIPS 2020.
>
> [2] Wang, A.,  et al. HINT: Hierarchical Neuron Concept Explainer. CVPR  2022.
>
> [3]  Anh Nguyen, et al. Multifaceted feature visualization: Uncovering the different types of features learned by each neuron in deep neural networks. arXiv 2016.
>
> [4] Laura O’Mahony, et al. Disentangling neuron representations with concept vectors. CVPR 2023.
>
> [5] Jesse Mu et al. Compositional explanations of neurons. NeurIPS 2020
>
> [6] Bykov, Kirill, et al. "Labeling neural representations with inverse recognition." NeurIPS 2024.
>
> [7] Nick Cammarata et al. Thread: Circuits. Distill, 2020.

---

> > ### Author Response · Authors · 2024-11-22
> > **Follow-up**
> >
> > Dear Reviewer **qMvj,**
> >
> > We hope our response has thoroughly addressed your questions.
> >
> > We would greatly appreciate any further feedback or additional questions you may have.
> >
> > Please let us know.
> >
> > Thank you once again for your thoughtful insights and consideration.
> >
> > Best regards,
> >
> > Authors

---

> ### Comment · Reviewer_qMvj · 2024-11-22
>
> I would like to thank the authors for their detailed response.
>
> Since ICLR permits updating submissions, **I strongly suggest the authors to upload an updated version of their manuscript to OpenReview, reflecting the improvements they intend to include**. Even if this is not the final camera-ready version, having the updated version available would greatly assist reviewers in evaluating the revisions and adjusting their scores accordingly.
>
>
> Unfortunately, my primary concerns remain unaddressed. Specifically, I had requested additional experimental validation and an **experimental comparison of different approaches**. Below, I will provide a better wording of these requests. While the conceptual differences were already sufficiently explained in the related work section, I appreciate the authors' effort to clarify them further.
>
>
> As stated in my initial review,
>
> *“The main weakness of the paper regards the evaluation and supporting experiments. The current evaluation adequately validates the framework’s design choices. However, **(the evaluation)** does not place the work in the broader context of the ongoing research on this topic.”*
>
> and
>
> *“several steps of the framework have been explored in prior literature, albeit sometimes with different objectives. Analyzing **(in terms of experimental evaluation)** the connections between the outcomes of these steps and findings from other approaches would greatly strengthen the paper.”*
>
> I have added words in parentheses to clarify the request. While I understand that alternative methods may differ slightly from the proposed approach, the purpose of explanation methods is to provide insights. Analyzing these differences in terms of experimental results would strengthen the contribution of the paper.
>
>  **I will use the response to R1 - (Q1 concern) as an example to better highlight my concerns. However, the same concern (and analysis) applies to Q3 and Q4. I am prioritizing speed over precision in this response and phase to give to the authors as much time as possible to address these issues**.
>
> R1:  I understand that the paper’s definition and methodology for extracting “critical neurons” differ somewhat from prior approaches. This distinction might even justify rebranding these neurons under a different name. However, the manuscript states:
> - *Focusing on classification models, **we construct, for each class of interest, a hierarchical tree** in which nodes represent critical neuron groups (defined by the concepts they encode), and edge weights quantify the interactions between these groups.*
> - *(Figure 7) **Using NeurFlow to reveal the reason behind model’s prediction**. The top concepts can be traced throughout the circuit*
> - *(Figure 8) Demonstration for automatically labelling and **explaining the relation of NGCs on class “great white shark”** using GPT4-o*
> - Line 150-152 use datasets composed of samples **classified by the model as a specific class c**
> - The visual features in 3.3 are computed based on the dataset built in the previous point
> and many other examples in the rest of the paper
>
> These examples indicate a relationship between the constructed tree (or circuit) and the model’s predictions. As I mentioned earlier, it would be beneficial to analyze experimentally how the proposed circuit/tree/neurons compare with those extracted by alternative methods. This was also suggested in other reviews. While I acknowledge that the proposed approach could theoretically build a tree from random neurons instead of output neurons, the paper’s narrative and its applications are grounded in the connection to predictions and classifications. Therefore, I believe it is important to either provide experimental evidence supporting this narrative or reconsider the framing of the manuscript.
>
> EDIT: The purpose of the additional experiments it's not proving that the proposed method is better than alternative approaches in a given task (e.g., identification of critical neurons and/or circuits) but to analyze and explain the differences between the authors' findings and the findings of alternative frameworks (e.g., they could be complementary or in contradiction with each other)

---

> ### Author Response · Authors · 2024-11-24
> **Additional experiments (part 1/3)**
>
> &nbsp;
> **We sincerely thank the Reviewer for taking the time to read our response and follow up on the discussion.
> Your valuable contributions will greatly help us improve our work.**
>
> Below are our responses to your concern.
>
> >**F1. I strongly suggest the authors to upload an updated version of their manuscript to OpenReview, reflecting the improvements they intend to include. Even if this is not the final camera-ready version, having the updated version available would greatly assist reviewers in evaluating the revisions and adjusting their scores accordingly.**
>
> **Response:** We thank the Reviewer for a thoughtful suggestion.
>
> We will incorporate all the new content into the Manuscript and Supplementary Materials and upload the updated versions to OpenReview.
>
> &nbsp;
>
> >**F2.  These examples indicate a relationship between the constructed tree (or circuit) and the model’s predictions. As I mentioned earlier, it would be beneficial to analyze experimentally how the proposed circuit/tree/neurons compare with those extracted by alternative methods. This was also suggested in other reviews. While I acknowledge that the proposed approach could theoretically build a tree from random neurons instead of output neurons, the paper’s narrative and its applications are grounded in the connection to predictions and classifications. Therefore, I believe it is important to either provide experimental evidence supporting this narrative or reconsider the framing of the manuscript.**
>
> >**The purpose of the additional experiments it's not proving that the proposed method is better than alternative approaches in a given task (e.g., identification of critical neurons and/or circuits) but to analyze and explain the differences between the authors' findings and the findings of alternative frameworks (e.g., they could be complementary or in contradiction with each other).**
>
> **Response:** We appreciate the Reviewer’s detailed comments and have made every effort to address the concerns by conducting additional experiments.
> Specifically, we performed **two quantitative experiments** and **one qualitative experiment** to compare the findings of our method with those of previous works [1] and [2].
> Unfortunately, we were unable to include comparisons with [4] due to its expensive computational cost (~5000x more forward passes than our approach).
>
> The details of the experiments are outlined below:
>
> **Experiment 1: Qualitative Analysis**
> This experiment addresses:
> >**F1**: "analyze and explain the differences between the authors' findings and the findings of alternative frameworks?"
> >**Q1**: “How do the findings in these papers relate to those in this work?”
>
> We compared our method with NeuCEPT [1] in identifying critical neurons at layer 4.2 of ResNet50.
> Using the experiment setup described in “Image Debugging” (similar to Figure 9, Section 5.1 in our manuscript), we followed these steps:
> * Both methods were used to identify the top-2 groups of critical neurons for a given  misclassified image. Groups with the highest metric scores (defined in line 453) are selected.
> * The critical neurons in these groups were masked, and the resulting changes in prediction probability were observed.
>
> To ensure fairness, we selected three classes without cherry-picking: Bald Eagle, Great White Shark, and Bee (corresponding to Figures 7, 8, and 9 in our paper). Groups of neurons were identified following the methodology described in our manuscript.
>
> The results of this comparison are provided in the [Link](https://anonymous.4open.science/r/NeurFlow-Rebuttal-images-3F6E/R3_Q1_3_4_NeurFlow_rebuttal.pdf).
>
> Qualitatively, we observed that our method identified the top-2 concepts more closely resembling the original images.
>
> Additionally, our *top logit drop images* (i.e., "images showing the largest decrease in the target logit value" as described on Line 485 of our paper) better matched the representative examples of the identified concepts.
>
> Furthermore, masking the critical neuron groups identified by our method resulted in more significant changes to the prediction probabilities, using fewer neurons, compared to the groups identified by NeuCEPT [1].
> For instance, with the labels *Bald Eagle* and *Great White Shark*, masking NeuCEPT’s critical neurons had no effect on prediction probabilities, whereas masking the neurons identified by our method substantially altered the predictions.
>
> These findings suggest that our approach identifies more impactful neurons and concepts directly related to the model’s predictions compared to NeuCEPT.

---

> ### Author Response · Authors · 2024-11-24
> **Additional experiments (part 2/3)**
>
> **Experiment 2: Quantitative experiment concerning identify critical neuron to the output of the model**
>
> This experiment is to respond:
> >**Q1: “Is there an overlap between their critical neurons and those identified here? If not, what could explain the differences?”**
>
> We assess the degree of overlap between the critical neurons identified by our method and those identified by NeuCEPT. For this evaluation, we utilize the NeuronMCT skyline [3], which is specifically designed to identify neurons critical to the model’s output, as recommended by Reviewer 4.
> The more overlap with the skyline, the more “critical” the neurons to the final prediction.
>
> We evaluated all layers of both models and measured the average **F1 scores** of the overlaps. We measure across 10 random classes:
>
> | Overlap between: | NeuronMCT -  **NeurFlow** | NeuronMCT - NeuCEPT | **NeurFlow**  - NeuCEPT |
> |:----------------:|:-------------------------:|:-------------------:|:-----------------------:|
> |     ResNet50     |            0.72           |         0.48        |           0.49          |
> |     GoogLeNet    |            0.79           |         0.55        |           0.56          |
>
> The results suggest that:
> * The neurons identified by our method align more closely with the NeuronMCT skyline, indicating that our critical neurons have a stronger impact on the model's output.
> * The overlap between neurons identified by our method and NeuCEPT is approaximately $50\\%$ in terms of F1-score.
>
> However, we were unable to conduct the “Fidelity of Critical Neurons” experiment due to the extensive runtime required by NeuCEPT.
>
> &nbsp;
>
> **Experiment 3: Quantitative experiment concerning identifying critical neurons of a specific target neuron**
>
> This experiment addresses:
> >**Q3:** “A large body of literature explores circuit extraction (as cited in the paper). Although circuit structures may vary, it would be insightful to compare their components. Do they share the same neurons?”
>
>
> To explore this, we compare the critical neurons of a specific target neuron identified using our proposed Integrated Gradient scores with those identified by [2]. In [2], neurons are ranked based on the top neurons with the highest $L_2$ weights connected to the target neuron. (Note that this method is not applicable in other experiments since calculating weight magnitude is limited to consecutive layers.)
>
> For this comparison, we identify the top $\tau$ critical neurons in two consecutive layers (separated by one convolution layer, as per the setup in [2]) using both methods. We then knock out these critical neurons to observe how the target neuron’s concept is affected.
>
> The extent of this change is quantified by the loss function defined on Line 348 (experiment “Optimality of critical neurons”), where a lower loss indicates better performance. We randomly selected 100 neurons across 10 different convolution layers from both models and calculated the average difference in losses between the two methods. A negative result indicates our method produces a better loss, while a positive result indicates otherwise.
>
> The findings are summarized below:
>
> |   Model   | Average subtraction of the losses  (negative means our loss is better and vice versa) |
> |:---------:|:-------------------------------------------------------------------------------------:|
> | ResNet50  | -0.082                                                                                |
> | GoogleNet | -0.013                                                                                |
>
> These findings demonstrate that our method is more effective at identifying critical neurons. Additionally, gradient-based approaches are more versatile, as they can be applied to non-consecutive layers (e.g., ResNet Block 4.2 → ResNet Block 4.1 in our experiments), whereas the $L_2$-weight-based approach is limited to consecutive layers.
>
> &nbsp;
>
> **References:**
>
> [1] Vu, Minh N., et al. "NeuCEPT: Locally Discover Neural Networks' Mechanism via Critical Neurons Identification with Precision Guarantee." arXiv 2022.
>
> [2] Nick Cammarata, et al. Thread: Circuits. Distill, 2020.
>
> [3] Khakzar, Ashkan, et al. "Neural response interpretation through the lens of critical pathways." CVPR 2021.
>
> [4] Amirata Ghorbani et al. Neuron shapley: discovering the responsible neurons. NeurIPS 2020.
>
> [5] Wang, A., et al. HINT: Hierarchical Neuron Concept Explainer. CVPR 2022.
>
> [6] George A Miller. Wordnet: a lexical database for english. Communications of the ACM,1995.

---

> > ### Author Response · Authors · 2024-11-24
> > **Additional experiments (part 3/3)**
> >
> > **Regarding Question 4:**
> >
> > We were unable to directly compare our method with [5] due to a key challenge: **our approach** groups neurons based on **low-level concepts** (e.g., shark heads, teeth), whereas **their method** is focused on **high-level concepts** (e.g., animals, people). **A direct comparison would require manually labeling the dataset with low-level concepts** or using the WordNet [6] dataset from their setup, which is unrelated to one-class classification and beyond the scope of this study.
> >
> > We hope our responses have clarified the Reviewer’s concerns.
> > If you have any further questions, please let us know.
> > We would be happy to continue the discussion.

---

> ### Author Response · Authors · 2024-11-28
> **We have uploaded the revised Manuscript and Appendix**
>
> Dear Reviewer **qMvj**,
>
> We have made substantial revisions to the main text and appendix to address the Reviewers' feedback.
>
> The updated versions have been uploaded to OpenReview.
>
>
> Below is a detailed summary of the revisions we have made to address your concerns:
>
> 1. We replaced the term **Critical neuron** with **Core concept neuron** to more accurately convey the role of these neurons (i.e., their significance in encoding concepts).
> This revision also avoids potential confusion with existing "critical neurons" in the literature, emphasizing the conceptual novelty of our work (e.g., **Lines 042-045; Lines 060-061**).
>
> 2. We revised the Introduction  and abstract to highlight the distinction between our approach and existing methods. In particular, we now emphasize our focus on exploring inter-layer interactions of neuron groups, as opposed to examining the relationship between individual neurons and the model's output.
>
> 3. We conducted additional experiments to highlight the distinctions between our method and others, including:
>       * Qualitative comparisons of core concept neurons identified by NeurFlow versus critical neurons detected by other methods (**Lines 502-508; Appendix D.3**).
>       * Quantitative comparisons of NeurFlow with existing approaches (**Lines 419-431; Appendices D.4, D.5**)
>       * An assessment of the completeness of core concept neurons (**Lines 392-399; Appendix D.6**)
>
> 4. A new discussion section (**Appendix C**) has been added to address the limitations of our proposed method, as per your suggestion.
>
> 5. We revised our terminology, using "patches" where appropriate and reserving "visual features" to refer specifically to their semantic meanings (e.g., **Lines 160-164; 200-205**)
>
> We sincerely hope these comprehensive revisions meet your expectations.
>
> If so, we would greatly appreciate your kind consideration of a potential score improvement.
>
> Please let us know if you have any further questions or require additional clarification.
>
> We would be delighted to continue the discussion.
>
> Authors

---

> ### Comment · Reviewer_qMvj · 2024-11-29
> **Summary of Review After Rebuttal**
>
> Thank you to the authors for their clarification and the detailed summary. In my opinion, the revised version of the paper is stronger than before, and I raised my score to 6 to reflect this improvement. While some concerns remain partially addressed (justifying my score), I understand there was limited time and the improvements made are substantial enough to lean towards acceptance. I encourage authors to include the remaining experiments in the camera-ready version.
>
>
> The area chair and authors can find below a summary of my previous concerns and solutions the authors provided to address such concerns. This section also includes some additional feedback for the authors for the camera ready version.
>
> - The paper did not include experiments comparing their findings with current literature. The only experiments were ablation studies. Specifically, I requested experiments against methods for critical neuron identification, circuits, and neuron grouping. The authors partially addressed these concerns by adding quantitative and qualitative experiments against a couple of methods for critical neuron identification. While there is still no comparison for neuron grouping and circuits, the novelty and the contribution of the paper could compensate for this limitation.
> - The paper lacked supporting experiments for the claim  “demonstrate that for a particular task, **only** a subset of neurons has a substantial influence on the model’s performance”. The authors provided additional experiments showing that “including more non-critical neurons into the set of critical neurons identified by our algorithm **does not significantly improve** the model's performance.” However, the usage of the word “only” remains too strong given the evidence (also considering the results for GoogleNet). I **recommend removing the word “only”** and rephrasing the sentence in something like “demonstrates that for a particular task, the identified core neurons have a substantial influence […]”
> - The paper did not discuss the limitations of some of its assumptions. The authors addressed this concern by adding a section to the appendix discussing the limitations of the proposed approach.
> - The terminology used was inconsistent and confusing in some sections. The authors fully addressed this concern by renaming some concepts and improving the clarity of the text.
>
> Additional feedback:
> - Please provide an explanation in the paper or additional experiments to justify some hyperparameter choices. For example, in the comparison with NeuCEPT, why do the authors choose to consider only the top 2 groups and not the top 1 or the top 3? If there is a technical explanation, please include it in the paper. If not, please add more experiments with several thresholds in the camera-ready version.
> - I suggest clearly stating the fact that the proposed approach groups neurons based on low-level concepts at the beginning of the paper. Currently, this is implicit and must be deducted.  Declaring this feature of the approach better places the proposed work in the literature and this information would be useful for potential downstream tasks. Note also that even though  [5] captures high-level concepts, it is still useful to compare qualitatively the approaches (e.g., a neuron that captures the concept of shark head should be associated with the shark or animal concept by [5] if the approaches agree on the concept assignment)

---

> > ### Author Response · Authors · 2024-11-29
> > **Thank you for your feedback and improving our score**
> >
> > Dear Reviewer **qMvj**,
> >
> > We deeply value the time and effort you have invested in reviewing our work and sharing your insightful feedback.
> >
> > Your input has been invaluable in enhancing the quality of our research.
> >
> > Thank you for kindly considering an adjustment to our score.
> >
> > Moving forward, we will make every effort to refine the paper by addressing the reviewers' additional comments. This includes revising specific terms to ensure clarity and precision, avoiding possible misinterpretations, and conducting further experiments to enhance the comprehensiveness of our work. All updates will be thoughtfully incorporated into the final version.
> >
> > Authors

---

### Official Review · Reviewer_fHh7 · 2024-11-04

**Soundness:** 3
**Presentation:** 2
**Contribution:** 3
**Rating:** 6
**Confidence:** 3

**Summary:**

This paper argues that it is most useful to analyze neural networks through the lens of groups of neurons rather than individual neurons. It describes an algorithm for constructing such groups, which can be used to construct hypertrees of which neuron groups are involved in each class prediction. The algorithm works by creating a dataset of visual features (which I believe are different crops of different dataset classes), then for each neuron constructing the "neuron concept", which is the top K visual features that the neuron activates. The algorithm then looks for the critical input neurons to each neuron, which is defined as ones which would change the neuron concept if removed. (This is actually done via an importance score, since calculating this fully in set form would be too computationally costly). Then, one can construct a hypertree of neurons involved in predicting a class, based on finding the critical neurons of the final logit and then on downwards.

**Strengths:**

- This paper engages with the problem of neuron analysis in a way that seems to pragmatically balance between the purist but unscalable approach of individual neurons, and the combinatoric explosion of looking at every possible circuit
- The paper did rigorous analysis of both the correctness and the usefulness for interpretability of the neurons they find, as well as calling out explicit useful applications

**Weaknesses:**

The paper defines a lot of terms (critical neuron group, neuron concept, neuron circuit/hypertree, visual feature) that are in colloquial language somewhat similar, and I think it could have benefitted from a glossary figure that explicitly defines all of these with reference to one another, so that a reader can flip to it if needed

**Questions:**

- I was unclear on how the semantic grouping vector was calculated, and which feature maps were explicitly within each visual feature
- Are visual features individual images? Images of a class? Some subset of images within a class? I was unclear on this, and it made it harder to understand the rest of the paper

---

> ### Author Response · Authors · 2024-11-20
> **Response to Reviewer fHh7**
>
> &nbsp;
>
> **We sincerely thank the Reviewer for very constructive comments.
> We provide our responses to the Reviewer's feedback below.**
>
>
> >**Q1. The paper defines a lot of terms, the reviewer suggests that the authors should create a glossary figure that explicitly defines all of the reference terms.**
>
>
>
> **Response:** We thank the Reviewer for a thoughtful comment.
> We will add a glossary figure into the Supplementary of the final version.
>
>
> &nbsp;
>
>
> >**Q2. I was unclear on how the semantic grouping vector was calculated, and which feature maps were explicitly within each visual feature**
>
> **Response:** The algorithm for calculating the semantic grouping vector is detailed in Appendix B.2 (Lines 815-818), as referenced in Lines 292-293. The semantic group vector is formed by averaging the representative vectors of the visual features within the semantic group, which includes all feature maps of the layer.
>
> &nbsp;
>
>
> >**Q3. What are the visual features? (individual images, images of a class, a subset of images within a class)**
>
> **Response:** In short, visual features are crops of images.
>
> As stated in Lines 166-168, we enhance the original dataset $\mathcal{D}_c$ (corresponding to a specific class $c$) by partitioning it into smaller pieces of varying sizes, where smaller patches capture simpler visual features and larger patches represent more complex ones (Lines 203-204). We use these augmented samples as visual features to probe the model.
>
> &nbsp;
>
> **If our explanations have addressed the Reviewer's questions, we kindly hope the Reviewer might consider increasing our score.**
>
> **Should there be any remaining issues or points of concern, please let us know.
> We will gladly follow up to provide further clarification.**

---

> > ### Comment · Reviewer_fHh7 · 2024-11-26
> >
> > **On visual features:** I think that "partitioning [a dataset] into smaller features of various sizes" does not make it clear that what is taking place is cropping. (I had been thinking that "partitioning a dataset into smaller size" meant creating a sub-dataset with fewer examples. This confusing wording seems to be still present in the final PDF.
> >
> >
> > **On semantic group representation:** While I appreciated the reference to the location in the appendix where this calculation is described, I think it (1) should be in the main text of the paper, and (2) would benefit from an in-words description of the calculation being performed, to supplement and provide context for the mathematical one.
> >
> > On the whole, and given no substantial revision to the paper as it currently stands on OR, I'm keeping my score as-is.

---

> > > ### Author Response · Authors · 2024-11-27
> > > **Thank for the feedback**
> > >
> > > We would like to note that **we have not yet uploaded the revised manuscript and supplementary materials.** Currently, we are incorporating the content from discussions with the Reviewers into the manuscript and supplementary materials and **will upload them before the deadline (December 27)**. At that time, we hope you will kindly reevaluate our revised submission in light of the updates and improvements we are making. Thank you for your understanding and thoughtful feedback throughout the review process.
> > >
> > > &nbsp;
> > >
> > > **Below are our responses to your concerns:**
> > >
> > >
> > > >**F1. On visual features: I think that "partitioning [a dataset] into smaller features of various sizes" does not make it clear that what is taking place is cropping. (I had been thinking that "partitioning a dataset into smaller size" meant creating a sub-dataset with fewer examples. This confusing wording seems to be still present in the final PDF.**
> > >
> > > **Response:** We thank the Reviewer for the feedback.
> > > To clarify, we have revised the sentence to “Instead, we enhance the original dataset $\mathcal{D}_c$ by **cutting it into smaller patches** with varying sizes. These patches serve as visual features for probing the model.”
> > >
> > > &nbsp;
> > >
> > > >**F2. On semantic group representation: While I appreciated the reference to the location in the appendix where this calculation is described, I think it (1) should be in the main text of the paper, and (2) would benefit from an in-words description of the calculation being performed, to supplement and provide context for the mathematical one. On the whole, and given no substantial revision to the paper as it currently stands on OR, I'm keeping my score as-is.**
> > >
> > > **Response:** We thank the Reviewer for the constructive comment.
> > > We will move the description from the Appendix to the main text as suggested.
> > >
> > > &nbsp;
> > >
> > > We hope our responses have addressed your concerns.
> > > Please let us know if you have any further comments or suggestions regarding our paper.

---

> > > > ### Author Response · Authors · 2024-11-28
> > > > **We have made substantial revisions to both the main text and the appendix.**
> > > >
> > > > Dear Reviewer **fHh7**,
> > > >
> > > > We have made significant revisions to both the manuscript and appendix to address the concerns raised by the reviewers.
> > > > The updated manuscript and appendix have been uploaded to OpenReview.
> > > >
> > > > Below are the updates we have made to address your concerns:
> > > >
> > > > * We added a table of notations in **Appendix A** to summarize all the symbols used in our paper.
> > > > * We included an explanation of the calculation of the *semantic grouping vector* in the main text (**Lines 289–290**).
> > > > * We revised the explanation of *visual features* to make it clearer and avoid potential misunderstandings (**Lines 160–161**).
> > > >
> > > > We would greatly appreciate if you could take some time to review the updated manuscript and appendix.
> > > >
> > > > We hope these updates effectively address the issues you raised.
> > > >
> > > > If that is the case, we would be sincerely grateful if you could consider revising your score for our paper.
> > > >
> > > > Please let us know if you have any further concerns.
> > > >
> > > > We would be more than happy to continue the discuss.

---

> > > ### Author Response · Authors · 2024-12-02
> > > **Could you review our revised manuscript and appendix?**
> > >
> > > Dear Reviewer **fHh7**,
> > >
> > > As the rebuttal period is nearing its conclusion, we would greatly appreciate it if you could kindly review our revised manuscript and appendix.
> > >
> > > We have made every effort to address your concerns in this revision and sincerely hope it meets your expectations.
> > >
> > > Should there be any remaining concern, please let us know.
> > >
> > > Thank you very much for your time and thoughtful feedback.
> > >
> > > Authors,

---

> ### Author Response · Authors · 2024-11-22
> **Follow-up**
>
> Dear Reviewer **fHh7,**
>
> We hope our response has thoroughly addressed your questions.
>
> We would greatly appreciate any further feedback or additional questions you may have.
>
> Please let us know.
>
> Thank you once again for your thoughtful insights and consideration.
>
> Best regards,
>
> Authors

---

### Official Review · Reviewer_gfeH · 2024-11-04

**Soundness:** 3
**Presentation:** 2
**Contribution:** 3
**Rating:** 8
**Confidence:** 3

**Summary:**

Summary:
The presented manuscript  proposes  the framework NeurFlow which offer interpretability to neural networks by focusing on groups of critical neurons. The framework tries to detect critical neurons and clusters them based on shared functionality, constructing a hierarchical circuit to model their interactions. The importance of the selected neurons is validated by comparing their impact on the model’s performance to random groups of neurons. Finally the manuscript also offers potential applications, e.g to debug model biases.

**Strengths:**

Strengths:
•	The identification of groups of critical neurons offers a way to tackle a node’s polysemantic behaviour.

•	The application of MLLMs to annotate group circuits is novel and original. This offers a nice way to explain a model’s decision.

•	The papers is very complete and self-contained. The proposed methodology is tested on different level as the authors don’t only look at the dependence of the model’s performance on the critical neurons, but also assess the optimality of their selection with the additional “Optimality of critical Neurons” experiment. Additionally the authors explore applications of their proposed approach.

**Weaknesses:**

Weaknesses:
•	The formal definition of a  group of critical neurons of is not obvious to me. Heuristically critical neurons are defined as neurons who, if knocked out, would result in a substantial change in performance. The formal definition however defines the critical neurons of neuron a as the group of neurons who, when knocked out, cause the neuron concept of neuron a ($\mathrm{V}^{\mathbb{S}}_a$) to be the most different from the original neuron concept ($\mathrm{V}_a$), as the cardinality of the intersection will then be small.

•	The description of the experiments are difficult to understand and it misses some important aspects to properly assess the results ( see questions below). The conclusion taken from these experiments are therefore not necessarily supported by the experiments.

**Questions:**

Questions:
1)	How dependent are the results on the choice of the hyperparameter k in the neuron concept $\mathrm{V}_a$ ?
2)	In the “neuron clustering algorithm based on the semantic groups”, is the concept of the neuron $s_i$, namely $\mathrm{V}_{s_i}$ a subset of the parent neuron’s concept $\mathrm{V}_a$? Intuitively, if the neurons are indeed critical for neuron a this should be the case correct? From Algorithm 3 in supplemental material it seems that the neuron concept $\mathrm{V}__{s_i}$ is identified independently of the parent’s concept.
3)	In the experiment to detect the optimality of the critical neurons the authors mention: “For each setting we randomly select 50 target neurons (denoted by a_i) from 10 distinct classes”. Does this mean the authors select in total 500 (50 x 10 neurons) or are 5 neuron selected per class ?
4)	In the experiments leading to Figure 4: Could the authors please indicate the confidence intervals as the results are averages over different selections of random groups of neurons ? It would be interesting to see the spread of the loss to note if some random groups of neurons reach near optimality.
5)	The weight $W(G_i,G_j)$ in Equation 5 seems to be unnormalized with respect to the number of neurons in $\mathbb{S_i}$ and $\mathbb{S_j}$. Doesn’t this make it difficult to interpret these weights over different groups of neurons?
6)	How are the random neurons selected in the masking/retaining experiment? Are the number of random neurons per layer selected in an equivalent amount as the number of critical neurons at that layer? Or is only the total number of random neurons matched to the cardinality of the set $\mathbb{S}_a$ ?
7)	The behaviour of the critical nodes in the multi-layer retaining experiment does not  seem to support the interpretation of critical nodes, as the performance significantly decreases (or fluctuates) with increasing layers.

Additional comments
Line 056: “there” -> “their”
Line 181: “our” -> “out”
Line 250: “its all” should be replaced with “all its”
Line 311: There seems to be a word missing from this sentence to make it coherent.
Figure 4: The current choice of color & marker shape is not very intuitive to grasp the message of the figure. Using a consistent colour for critical/random and varying the marker between different values of $\tau$ might make interpreting the figure more easy.

---

> ### Author Response · Authors · 2024-11-20
> **Response to Reviewer gfeH**
>
> &nbsp;
>
> **We sincerely appreciate the Reviewer’s detailed and thoughtful feedback.
> Please find our responses below.**
>
> >**Q1. The formal definition of a group of critical neurons of is not obvious to me. Heuristically “critical neurons are defined as neurons who, if knocked out, would result in a substantial change in performance.” The formal definition however defines the critical neurons of neuron a as “the group of neurons who, when knocked out, cause the neuron concept of neuron $V_{S^a}$ to be the most different from the original neuron concept $V_a$, as the cardinality of the intersection will then be small.”**
>
> **Response:** Indeed, we focus on the **inter-layer relationships** between neuron groups rather than the relationships between individual neurons and the model's final prediction results.
>
> As a result, our formal definition of a critical neuron refers to neurons in the preceding layer that are considered critical for a neuron in the immediately subsequent layer.
>
> Recursively, the set of all critical neurons identified as we trace backward from the final layer (the layer producing the prediction results) to the first layer constitutes the neurons that have the most significant influence on the model's prediction results.
>
> &nbsp;
>
> >**Q2. How dependent are the results on the choice of the hyperparameter $k$ in the neuron concept $\mathrm{V}_a$?**
>
> **Response:** To evaluate the dependence of the results on the choice of $k$, we conducted additional experiments with various values of $k$ and measured the number of critical neurons overlapping with the baseline setup of $k=50$. Greater overlap indicates less dependence on the choice of $k$.
>
> Table 1 summarizes the results with $\tau=16$ (i.e., the maximum number of critical neurons per target neuron is 16) and $k \in [ 30, 40, 50, 60, 70, 90, 110, 130, 150, 170, 190 ] $, evaluated across 50 random neurons.  The results show that for all tested values of $k$, the overlap ratio is always at least $14/16$ $( > 86 \\% ) $, demonstrating that the results of our proposed algorithm are independent of the choice of $k$.
>
> | K         | 30   | 40   | **50**   | 60   | 70   | 90   | 110  | 130  | 150  | 170  | 190  |
> |-----------|------|------|----------|------|------|------|------|------|------|------|------|
> | GoogLeNet | 15.0 | 15.4 | **16.0** | 15.5 | 15.3 | 15.3 | 15.0 | 15.0 | 14.9 | 14.9 | 14.9 |
> | ResNet50  | 14.9 | 15.6 | **16.0** | 15.6 | 15.3 | 14.7 | 14.5 | 14.3 | 14.1 | 14.0 | 14.0 |
>
> Table 1. Average number of overlapping critical neurons with various values of $k$.
>
>
> &nbsp;
>
> >**Q3. In the neuron clustering algorithm based on the semantic group, is the concept of the neuron $s_i$, namely $V_{s_i}$, a subset of the parent neuron’s concept $V_a$? Intuitively, if the neuron $s_i$ is indeed critical for neuron $a$, this should be the case correct?**
>
> **Response:** The set $V_{s_i}$ does not always need to be a subset of $V_a$.
> Instead, the concept **$V_a$** associated with the parent neuron $a$ can be understood as a combination of the concepts represented by its child neurons $V_{s_i}$ (as described in [1, 2]).
> This suggests that neuron **$a$** activates only for images that contain a specific **"combination" of the concepts** from its child neurons, which may result in low activation for the set $V_{s_i}$.
>
> For example, consider Figure 8: *Group 4* represents *"white skin and teeth,"*  while *Group 5* corresponds to *"eyes and an ocean background."* Meanwhile, *Group 2, the parent of Groups 4 and 5*, activates only images of *"shark heads with teeth and eyes"*. These images **would not include those featuring only "eyes" or "teeth."**
>
> &nbsp;
>
> >**Q4. In the experiment to detect the optimality of the critical neurons, the author mentioned: “For each setting we randomly select 50 target neurons from 10 distinct classes”. Does this mean the authors select in total 500 neurons or are 5 neurons selected per class?**
>
> **Response:** We select five neurons for each class and repeat this process across ten different classes, resulting in a total of 50 target neurons. In the final version, we will revise the sentence to ensure this setup is clearly understood.
>
> &nbsp;
>
> >**Q5. In the experiments leading to Figure4: Could the author please indicate the confidence interval as the results average over different selections of random groups of neurons?**
>
> **Response:**  We appreciate the Reviewer's insightful comment.
> We present the $99\\%$ confidence interval based on $5000$ runs for each column at [Link](https://anonymous.4open.science/r/NeurFlow-Rebuttal-images-3F6E/Confidence%20interval%20of%20fig%204.png).
> As illustrated in the figure, the confidence interval is minimal, highlighting the consistency of our reported results.

---

> ### Author Response · Authors · 2024-11-20
> **Response to Reviewer gfeH (con't)**
>
> >**Q6. The weight $W(G_i,G_j)$ in equation 5 seems to be unnormalized with respect to the number of neurons in $S_i$ and $S_j$. Doesn’t this make it difficult to interpret these weights over different groups of neurons?**
>
> **Response:** Note that each node (e.g., $G_i$, $G_j$)  in our concept circuit is associated with a group of concepts. Therefore, the relationship between $G_i$ and $G_j$ (quantified by $W(G_i, G_j)$) is represented by two aspects: the number of edges connecting elements of $G_i$ and $G_j$, and the weights of those connecting edges. The more the edges and the higher the edge weights, the stronger the relationship between $G_i$ and $G_j$. Thus, normalization in equation (5), $W(G_i, G_j)$ will only reflect the average magnitude of the edge weights without accounting for the number of edges between $G_i$ and $G_j$.
>
> &nbsp;
>
>
> >**Q7. In the masking/ retaining experiment, are the number of random neurons per layer selected in an equivalent amount as the number of critical neurons at that layer?**
>
> **Response:** The number of random neurons per layer is equivalent to the number of critical neurons at that layer. In the final version of the manuscript, we will clarify this setting.
>
> &nbsp;
>
>
> >**Q8. The behavior of the critical nodes in the multi-layer retaining experiment does not seem to support the interpretation of critical nodes, as the performance significantly decreases (or fluctuates) with increasing layers.**
>
> **Response:**  In the “multilayer retaining” setup, we mask out all non-critical neurons from the last layer to a specified layer.
> Hence, as we mask more layers, the model’s performance tends to drop or become unstable.
>
> Moreover, as we described in Lines 188-189, “the value of $\tau$ (i.e., the number of critical neurons) may vary across the network layers.” Indeed, the choice of the parameter $\tau$ is a tradeoff between the simplicity of the circuit and the completeness of capturing the critical neurons. The smaller the value of $\tau$, the more unstable the model's performance in the retaining experiment, and the greater the performance drop.
>
> With a reasonably chosen value of $\tau$ (large enough), for example, $\tau = 16$, the retaining experiment with critical neurons shows relatively stable performance, even when we increase the number of layers. This is most clearly observed in the multi-layer retaining experiment with $\tau = 16$, conducted on Resnet.
>
>
> &nbsp;
>
> **If our explanations have addressed the Reviewer's questions, we kindly hope the Reviewer might consider increasing the score.**
>
> **Should there be any remaining issues or points of concern, please let us know.
> We will gladly follow up to provide further clarification.**
>
>
> &nbsp;
>
>
> **References:**
>
> [1] Bykov, Kirill, et al. "Labeling neural representations with inverse recognition." Advances in Neural Information Processing Systems 36 (2024).
>
> [2] Nick Cammarata, Shan Carter, Gabriel Goh, Chris Olah, Michael Petrov, Ludwig Schu bert, ChelseaVoss, Ben Egan, and SweeKiat Lim. Thread: Circuits. Distill, 2020.

---

> > ### Comment · Reviewer_gfeH · 2024-11-24
> >
> > I want to thank the authors for their detailed response. The additional robustness checks, together with the clarifications, which can be included in the final manuscript are definitely positive.
> > I do have some small additional questions on two points the authors raise:
> > Q6.
> > While I understand the mathematical formulation of edge weights, I was wondering more on the aspect of how to compare $W(G_i,G_j$) between different $(G_i,G_j)$ combinations. Two groups of neuron with a lot of edges with small weights, could have the same branch weight as a group of neuron with less but stronger connections. Is the interpretation then the same for these equal branch weights?
> > Q8.
> > I appreciate the clarifications from the authors, however I wonder if they could further elaborate on the results for GoogLeNet. In this case we see a large fluctuation for $\tau = 8$, or we see a performance close to the performance of retaining random neurons for $\tau = 4$. Both these behaviours are very different from the behaviour for the same parameter values for ResNet. Does this imply a large dependence on the modeltype of this approach?
> > I appreciate the efforts from the authors and would gladly raise my score if both these final concerns are addressed.

---

> > ### Comment · Reviewer_gfeH · 2024-11-27
> >
> > I would like to thank the authors for their response and clarifications. As this discussion has addressed all the questions and the authors have made the necessary changes, I have decided to update my score to a 7.

---

> ### Author Response · Authors · 2024-11-22
> **Follow-up**
>
> Dear Reviewer **gfeH,**
>
> We hope our response has thoroughly addressed your questions.
> We would greatly appreciate any further feedback or additional questions you may have.
> Please let us know.
>
> Thank you once again for your thoughtful insights and consideration.
>
> Best regards,
>
> Authors

---

> ### Author Response · Authors · 2024-11-25
> **Follow-up**
>
> **We appreciate the Reviewer’s continued engagement and would like to address your feedback as below.**
>
> >**F1. While I understand the mathematical formulation of edge weights, I was wondering more on the aspect of how to compare $W(G_i,G_j)$ between different $(G_i,G_j)$ combinations. Two groups of neurons with a lot of edges with small weights, could have the same branch weight as a group of neurons with less but stronger connections. Is the interpretation then the same for these equal branch weights?**
>
> **Response:** We conducted an additional experiment to verify that: *groups of neurons with higher sum of scores will have higher impact on a target neuron, regardless of the number of neurons in the group.*
>
> We randomly sample $500$ groups of neurons of varying sizes, ranging from $[1, 5, 10, 20, 50]$. For a target neuron in the upper layer, we analyzed the correlation between the loss function (as defined in Line 348) and two metrics: **the average edge weights** within each group and **our original scoring method, which sums the edge weights** of neurons in the group.
> Higher absolute correlation values indicate a more effective scoring method.
>
> Due to time constraints, we evaluated only 10 randomly selected neurons with different labels in GoogLeNet.
>
> The results in [Link](https://anonymous.4open.science/r/NeurFlow-Rebuttal-images-3F6E/Correlation%20googlenet%20sum%20vs%20avg.png) demonstrate that our original scoring method achieved significantly higher correlations compared to the averaging approach. This suggests that the number of connections plays a key role in accurately measuring edge weights between neuron groups.
>
> &nbsp;
>
> >**F2. I appreciate the clarifications from the authors, however I wonder if they could further elaborate on the results for GoogLeNet. In this case we see a large fluctuation for $\tau=8$, or we see a performance close to the performance of retaining random neurons for $\tau=4$. Both these behaviors are very different from the behavior for the same parameter values for ResNet. Does this imply a large dependence on the model type of this approach?**
>
> **Response:** Empirically, we found that the effects of small values of $\tau$ (e.g., $\tau = 4, 8$) depend not only on the layers within a network but also on the specific network architecture. However, as $\tau$ becomes sufficiently large, its influence on layers or model types diminishes. For GoogLeNet, we hypothesize that the sensitivity to smaller $\tau$ values might be linked to the lack of skip connections in its architecture.
>
> To explore this further, we conducted additional experiments by increasing the number of critical neurons per target neuron in layer 5b to $20$ and $24$. The results, detailed in [Link](https://anonymous.4open.science/r/NeurFlow-Rebuttal-images-3F6E/Googlenet%20fidelity%20tau%2020%2024.png), show that with these higher $\tau$ values, the performance drop of the model becomes negligible. Furthermore, the differences between retaining $20$ or $24$ neurons at layer 5b are minimal, suggesting that the dependence on $\tau$ decreases at this level.
>
> &nbsp;
>
> We hope our responses have satisfactorily addressed your concerns.
>
> If you have any further questions, please let us know.
>
> We would be delighted to continue the discussion.

---

> ### Author Response · Authors · 2024-11-28
> **Thank you very much for improving our score**
>
> Dear Reviewer **gfeH**,
>
> We sincerely thank you for taking the time to discuss with us, provide valuable feedback to improve our paper, and especially for considering revising our score upward.
>
> We have uploaded an updated manuscript and appendix to the system, incorporating the contents discussed with the reviewers.
>
> Below is a summary of the revisions we made to address the concerns and suggestions you raised:
>
> **1. Terminology Update:**
>    * We replaced the term **"Critical neuron"** with **"Core concept neuron"** to more accurately convey the role of these neurons (i.e., their significance in encoding concepts). This revision also avoids potential confusion with existing "critical neurons" in the literature, emphasizing the conceptual novelty of our work (e.g., **Lines 042-045; Lines 060-061**).
>
> **2. Additional Experiments:**
> We conducted several additional experiments, including:
>    * Evaluating the impact of the parameter $k$ (**Lines 168-169; Appendix D.7**).
>    * Adding confidence intervals to the results of the Optimality of Core Concept Neurons experiment (**Figure 4**).
>    * Comparing our defined edge weight (via summation) to an alternative approach using averaging (**Appendix D.2**).
>    * Analyzing the impact of the parameter $\tau$ (**Lines 372-375; 393-399; Appendix D.6**).
>
> **3. Additional Explanations:**
>    * Clarifying the rationale behind our proposed edge weight (**Lines 310-314**).
>    * Providing explanations for the experimental settings of Optimality of Core Concept Neurons (**Lines 344-345**) and the masking/retaining experiment (**Line 360**).
>
>
> We hope these revisions comprehensively address your concerns.
>
> We would greatly appreciate it if you could update our score at your earliest convenience.
>
> Should you have any further questions or suggestions, please let us know.
>
> We would be delighted to continue discussing our work with you.

---

### Author Response · Authors · 2024-11-20
**Rebuttal by Authors**

We would like to thank the AC for securing four reviews with high quality.
We thank **Reviewer fHh7** for the accepting score.
We thank **Reviewers gfeH, qMvj and KddZ** for the detailed questions and very thoughtful comments, which help us highlight better the key contribution of our work.

Our responses to the Reviewers’ main questions are summarized below.
1. We distinguish the definition of *"critical neurons"* in our framework from those in the existing literature to highlight the conceptual novelty of our work **(Reviewers  gfeH, qMvj and KddZ)**.

2. We highlight the uniqueness of the circuit identified by our framework to showcase its contribution **(Reviewers qMvj, KddZ)**.

3. We perform additional experiments to validate the efficiency of our framework **(Reviewers gfeH, qMvj, KddZ)**.

4. We offer a more comprehensive explanation of specific algorithms and notations used in our framework, we also provide rationale behind our algorithm **(Reviewers gfeH, fHh7, KddZ)**.

We welcome any follow-up questions from the reviewers regarding our rebuttal. We hope that, based on our detailed responses, the reviewers will consider increasing their scores if their concerns have been sufficiently addressed.

---

### Author Response · Authors · 2024-11-27
**Summary of the revised Manuscript and Supplementary Materials**

Dear AC and Reviewers,
We extend our sincere gratitude to the Reviewers for engaging in thoughtful discussions with us and providing valuable feedback. Your insights have been instrumental in improving our work.

We have carefully addressed all comments and suggestions from the Reviewers, ensuring that their feedback is thoroughly incorporated into both the revised manuscript and the Appendix.

All changes in the revised manuscript and supplementary materials are highlighted in blue.

We believe the updated submission reflects significant improvements, effectively clarifying and resolving the concerns previously raised.

&nbsp;

Below, we summarize the key changes made to our revised manuscript and supplementary:

1. **Terminology Update** (Reviewers **gfeH, qMvj, KddZ**):

* We replaced the term **Critical neuron** with **Core concept neuron** to more accurately convey the role of these neurons (i.e., their significance in encoding concepts). This revision also avoids potential confusion with existing "critical neurons" in the literature, emphasizing the conceptual novelty of our work.

2. **Introduction Revision** (Reviewer **qMvj**, **KddZ**):

* The Introduction has been revised to highlight the distinction between our approach and existing methods. In particular, we now emphasize our focus on exploring inter-layer interactions of neuron groups, as opposed to examining the relationship between individual neurons and the model's output.

3. **Additional Experiments:**

We performed several new experiments to better differentiate our approach from prior works, including both quantitative and qualitative evaluations. These experiments include:
* Analyzing the impact of the parameter $\tau$ (**Lines 372-375; 393-399; Appendix D.6**) (Reviewer **gfeH**),
* Examining the influence of the parameter $k$ (**Lines 168-169; Appendix D.7**) (Reviewer **gfeH**),
* Comparing our defined edge weight (by summing) to the one using averaging (**Appendix D.2**) (Reviewers **gfeH**),
* Conducting qualitative comparisons between the core concept neurons identified by NeurFlow and the critical neurons detected by other methods (**Lines 502-508; Appendix D.3**). (Reviewers **qMvj**, **KddZ**)
* Performing quantitative comparisons of NeurFlow with existing approaches (**Lines 419-431; Appendices D.4, D.5**) (Reviewer **qMvj**, **KddZ**),
* Evaluating the completeness of core concept neurons (**Line 392-399, Appendix D6**) (Reviewer **qMvj**)
* Adding confidence intervals to the Optimality of Core Concept Neurons experiment results (**Figure 4**) (Reviewer **gfeH**).
* Adding Shapley-based attribution method in our comparison of attribution method for edge weight (**Figures 10, 11; Appendix D.1**) (Reviewer **KddZ**)

4. **Enhanced Explanations:**

* To improve clarity, we incorporated additional explanations as suggested by all Reviewers (e.g., **Lines 160-161**  (Reviewer **fHh7** ), **Lines 289-290** (Reviewer **fHh7**), **Lines 310-313** (Reviewer **gfeH**), etc.).
* We also included a table of notations in **Appendix A** (as suggested by Reviewer **fHh7**).

5. **Inclusion of Relevant Works:**
* We have incorporated all relevant studies highlighted by the Reviewers into the revised manuscript (**Lines 094-106**) and provided a clear explanation of how our proposed method differs from existing approaches (**Lines 126-130**).

6. We add a discussion on limitations and future direction as suggested by Reviewers **KddZ**, **qMvj**.

&nbsp;

We hope all Reviewers are satisfied with our revised manuscript and appendix.

We would be deeply grateful if you could kindly reconsider and increase our score, given the substantial improvements we have made in this revised version.

---

### Meta-Review · Area_Chair_ts1b · 2024-12-19

**Metareview:**

The authors present NeurFlow, a novel framework designed to enhance the interpretability of neural networks by focusing on the interactions between groups of critical neurons. Unlike traditional methods that examine individual neurons, NeurFlow identifies core concept neurons: groups of neurons that collectively encode meaningful concepts and model their interactions across layers in the network. The paper introduces a hierarchical "concept circuit" that illustrates how these neuron groups contribute to the model's decision-making process.

The weaknesses of this paper include complex terminology, writing in the introduction, and insufficient experiments in quantitative and qualitative evaluations. However, during the rebuttal period, the authors actively engaged in discussions with the reviewers, and the final scores were (8, 6, 6, 6). All reviewers rated the paper positively.  Overall, this paper makes a valuable contribution to identifying critical neurons and addressing the polysemantic nature of neurons. It offers a fresh, well-structured perspective on interpretability, with promising, practical applications. So I recommend it for acceptance at ICLR.

**Additional Comments On Reviewer Discussion:**

The paper receives all positive scores, with one strong positive (8) and three borderline accept (6, 6, 6).

The authors have thoroughly addressed the reviewers' concerns, and these responses are reflected in the revised submission. The comprehensive improvements in the revised version have been well-received by the reviewers, including clearer terminology (Reviewers gfeH, qMvj, KddZ, fHh7), more detailed analysis and comparison with existing approaches (Reviewer qMvj, KddZ), and experimental validation of the framework's robustness and effectiveness (Reviewers gfeH, qMvj).

The improvements made during the rebuttal process have significantly enhanced the quality of the paper, and I recommend it for acceptance.

---

### Decision · Program_Chairs · 2025-01-22

Accept (Poster)